# Increased uptake of tuberculosis preventive therapy (TPT) among people living with HIV following the 100-days accelerated campaign: A retrospective review of routinely collected data at six urban public health facilities in Uganda

Joseph Musaazi[1]*, Christine Sekaggya-Wiltshire[1], Stephen Okoboi[1], Stella Zawedde-Muyanja[1], Mbazi Senkoro[2], Nelson Kalema[1], Paul Kavuma[1], Proscovia M. Namuwenge[3], Yukari C. Manabe[1,4], Barbara Castelnuovo[1], Agnes Kiragga[1]

1 Infectious Diseases Institute, Makerere University College of Health Sciences, Kampala, Uganda,
2 National Institute for Medical Research, Muhimbili Centre, Dar-es-Salaam, Tanzania, 3 National Tuberculosis and Leprosy Program, Uganda Ministry of Health, Kampala, Uganda, 4 Division of Infectious Diseases, Department of Medicine, Johns Hopkins University School of Medicine, Baltimore, Maryland, United States of America

* musaazijoseph7@gmail.com

## Abstract

Tuberculosis preventive therapy (TPT) effectively decreases rates of developing active tuberculosis disease in people living with HIV (PLHIV) who are at increased risk. The Uganda Ministry of Health launched a 100-day campaign to scale-up TPT in PLHIV in July 2019. We sought to examine the effect of the campaign on trends of TPT uptake and characteristics associated with TPT uptake and completion among persons in HIV care. We retrospectively reviewed routinely collected data from 2016 to 2019 at six urban public health facilities in Uganda. HIV care database and paper-based TPT registers at six public health facilities in Kampala, Uganda were retrospectively reviewed. Estimated trends of TPT (given as Isoniazid monotherapy) uptake and completion across the 4 years, among PLHIV aged 15 years and above, and factors associated, were examined using Poisson regression model with robust standard errors using generalized estimating equation (GEE) models. On average, a total of 39,774 PLHIV aged 15 years and above were eligible for TPT each calendar year at the six health facilities. Across all 4 years, more than 70% were females (range: 73.5% -74.6%) and the median age ranged from 33 to 34 years. From 2016 quarter one to 2019 quarter two, TPT uptake was consistently below 25%, but, as expected, the uptake significantly increased by about 3-folds from 22.1% to 61.2%, in 2019 quarter two (i.e. before the roll-out of the 100-day accelerated TPT intervention) and quarter three (i.e. after the roll-out of the 100-day accelerated TPT intervention) respectively. This increase remained highly significant even after adjusting for patients' baseline characteristics (adjusted prevalence ratio [aPR] = 2.58 [95%CI 2.45, 2.72], P-value<0.001). TPT completion was consistently high at above 70% at most of the time, but, it increased significantly

**Data Availability Statement:** All relevant data are within the paper and its Supporting Information files.

**Funding:** Support for data collection was provided by: 1) European & Developing Countries Clinical Trials Partnership (EDCTP) - East Africa TB NODE. Grant number: EDCTP-RegNET2015-1104 2) Fogarty International Center, National Institutes of Health (grant # 2D43TW009771-06 "HIV and co-infections in Uganda." The funders had no role in the study design, data collection, and analysis, decision to publish, or preparation of the manuscript.

**Competing interests:** The authors have declared that no competing interests exist.

among those initiated during 2018 quarter four and in the subsequent two quarters after the roll-out of the 100-day accelerated TPT intervention (i.e. TPT completion was: 83.2%, 95.3%, and 97.1% among individuals initiated during 2018 quarter4, and 2019 quarters 1 and 2, respectively). The increase in TPT completion during this period remained significant even after adjusting for patients' baseline characteristics (aPR [95%CI] = 1.09 [1.04, 1.14], P value<0.001, and 1.10 [1.05,1.15], P value<0.001, for individuals initiated during 2019 quarter 1, and 2, respectively compared to those initiated during 2018 quarter 4). Not on ART or newly started on ART compared to ART experienced, and pregnant at TPT initiation compared to not pregnant were associated with poor TPT completion, whereas older age ($\geq$25 years versus 15–24 years) was associated with higher TPT completion. The targeted 100-day campaign dramatically increased TPT uptake and completion among PLHIV suggesting a viable catch up strategy to meet WHO guidelines. Future analysis with additional years of data post 100-days TPT intervention is required to evaluate the sustainability of the observed gains.

## Introduction

Tuberculosis (TB) is the most frequent cause of Human Immunodeficiency Virus (HIV)-related deaths worldwide despite the wide availability of antiretroviral therapy (ART) [1]. TB preventive treatment (TPT) reduces the risk of developing active TB [2] and TB-associated mortality among people living with HIV (PLHIV) [3, 4]. Therefore, the World Health Organization (WHO) recommends incorporating TPT into routine HIV care for all adults, adolescents and children living with HIV without symptoms suggestive of TB or without active TB [5]. Data from clinical trials, observational studies and routine care programs have demonstrated reduced TB incidence due to TPT use [6–9], with a dramatic drop in TB incidence in PLHIV when TPT is administered concomitantly with ART [6, 9, 10]. However, TPT implementation has been slow, and compounded by suboptimal treatment completion rates (60–80%) in resource-limited setting countries (RLS) [4, 11, 12]. In the recent 2020 guidelines for TB treatment, the WHO recommended shorter rifamycin-containing TPT regimens, which are associated with better completion rates [5]. However, such regimens are not yet widely available and isoniazid mono-therapy for 6 months is still the most frequently used regimen in RLS.

Uganda is among the WHO's 30 high TB/HIV burden countries which contribute about 60% of the total TB/HIV burden globally. In 2019, approximately 40% (35,000/88,000) of the new and relapsed TB patients were living with HIV and 6.7% of PLHIV newly enrolled in care were diagnosed with TB [1]. Since 2006, the Uganda Ministry of Health (MoH) recommended that all PLHIV without symptoms suggestive of active TB disease receive TPT given as a one course treatment regardless of CD4 count, ART status, history of TB treatment, and pregnancy status [13]. In 2014, the ministry developed the first TPT guideline that enables health workers provide TPT to PLHIV, and children under five years who have had contact with person with active TB disease [14]. The latest TPT guideline that was rolled-out in 2021, also includes shorter rifamycin-containing TPT regimens [15]. Observational studies conducted before 2019 showed that TPT uptake among PLHIV in Uganda was estimated at about 17% [16], with only 50%- 60% of patients completing treatment [16, 17]. Contributing factors to poor TPT uptake in Uganda included: inadequate TPT supply in health facilities, frequent drug stock-outs, poor patient adherence, limited TPT knowledge by health workers, lack of confidence in

symptom-based TB screening alone, and fear of isoniazid resistance [11, 16]. In order to boost the TPT program, the Uganda MoH instituted a 100-day accelerated isoniazid preventive therapy (IPT) scale-up campaign that was launched on 3[rd] July 2019 [18]. This campaign aimed to enroll 300,000 PLHIV on TPT given as isoniazid monotherapy at 1947 ART sites and ensure 100% completion rate by 30th September 2019, for the individuals initiated on TPT in quarters: October–December 2018, and January–February 2019. However, there is scanty published data showing the trends of TPT uptake and completion, and the impact of the 2019 Ugandan MoH 100-day TPT scale-up campaign on these trends.

This study aimed to examine the effect of the 100-days campaign on trends of TPT uptake and completion from routinely collected data from 2016 to 2019 at six public health facilities in Uganda. The study also explored patients' characteristics associated with TPT uptake and completion among PLHIV in care.

## Methods

### Setting, study design and population

This was a retrospective cohort study based on review of patients' medical records at six of 10 HIV clinics in Kampala Capital City Authority (KCCA) public primary health care facilities (i.e., health centers) [19], which are supported by the Infectious Diseases Institute (IDI), Uganda. These six public health facilities were selected because their clinical and data management were supported by IDI personnel as a PEPFAR (President's Emergency Plan For AIDS Relief) implementing partner. Also, we selected HIV clinics in Kampala because it is one of the highest HIV prevalence areas in Uganda [20]; knowing its TPT program coverage and impact is crucial. Of the six health facilities, five were health center level III (Kiswa, Kawaala, Komamboga, Kitebi, Kisugu) and one was a health center IV (Kisenyi). Health center level IIIs have about 18 staff led by a senior clinical officer and provide: preventive, promotive, outpatient curative, maternity, inpatient health services, and laboratory services. Whereas health center level fours should have a senior medical officer and another doctor as well as a theatre for carrying out emergency operations in addition to all services provided at health center three [19]. All the selected six clinics were government health facilities providing free health care to about 2% of the population in Kampala city which approximated at 3.1 million in 2019 [21, 22], with 6.9% HIV prevalence among adults aged 15–64 year during 2016–2017 [20]. The bacteriologically confirmed TB prevalence was estimated at about 401 per 100,000 in urban areas during 2014–2015 [23]. Details of the activities provided at each health facility level are described in the Uganda health facility inventory report [19].

During the study period (January 2016 and December 2019), isoniazid monotherapy administered as a 6-month course was the only TPT option available, and was administered according to the 2014 Uganda national guidelines [14]. All adolescents and adults PLHIV who are unlikely to have active TB should receive at least a six-months course of isoniazid as TPT, regardless of immune status, ART status, history of TB treatment, and pregnancy status. Vitamin B6 (pyridoxine) was concomitantly given with isoniazid to prevent peripheral neuropathy [14]. Both TPT and vitamin B6 were provided to eligible individuals for free [14]. People newly enrolled into HIV care and those on ART refill visits are always re-evaluated for TPT history and screened for TB before initiation on TPT. Re-initiation of TPT is only done for anyone in HIV care who never completed or interrupted a course of TPT. In programmatic setting, depending on the stability of the patient in terms of health status and ART adherence, ART medications can be provided for either a month, 3 months, or 6 months. Therefore, TPT refill visits are aligned with ART refill visits in order to decrease the number of patient visits to the clinic which can be costly to the patient in terms of transport and time. In Uganda, TPT is

procured under the Quantification planning and procurement Unit (QPPU) of the ministry of health. Distribution of TPT is done by the government national warehouse (National Medical Stores i.e.NMS) while monitoring and support supervision is done by Ministry of Health through the National TB and Leprosy Program (NTLP).

In order to increase TPT coverage, on 3rd July 2019, the Ministry of Health launched a 100-day accelerated TPT campaign which aimed at enrolling 300,000 PLHIV on TPT given as isoniazid monotherapy and ensuring 100% completion rate by 30th September 2019 for the individuals initiated on TPT in quarters: October–December 2018, and January–February 2019 [18]. During the campaign there was: increased TPT availability at national level, improved TPT drugs access at health facilities, increased health workers' awareness campaigns on understanding the benefits of TPT, strengthening capacity of frontline workers in identification of eligible individuals, initiation, follow-up, and data management. Also, there was multi-stakeholder engagement for accountability on TPT, and engagement of civil society to empower clients and increase demand for improved TPT initiation and completion [18]. All the six health facilities studied were part of this campaign. The same national eligibility criteria for TPT initiation before the campaign, were maintained. The outline of activities and timelines for the 100-days campaign by the ministry of health and district leadership is presented in S2 Table X1 in S2 File.

## Inclusion and exclusion criteria

The study included all asymptomatic PLHIV aged 15 years and above, who were in care and had at least one clinic encounter from January 2016 to December 2019 at any of the six HIV clinics, who met the TPT eligibility criteria according to the national guidelines [14]. According to the guidelines, all PLHIV were eligible for TPT after screening for TB, i.e., without signs or symptoms of TB (i.e. cough of 24 hours or more, fever, weight loss or profuse night sweats), without active TB or not on TB treatment and without any contraindication to TPT. Individuals who were initiated on TPT in the previous calendar year were excluded in the analysis of subsequent years. Also, if a patient was eligible at start of the current year but during the course of the year he or she became ineligible e.g., diagnosed with TB, this patient could be considered in the denominator for the current year but excluded in denominators in the subsequent years. We included PLHIV aged 15 years and above because data was collected from adult HIV clinics and these enroll PLHIV aged 15 years and above in Ugandan setting. Also, we chose January 2016 as the starting point for our evaluation, because a health workers' guide for provision of isoniazid preventive therapy to PLHIV was rolled out in June 2014 and we wanted to allow enough time for the information to be disseminated and implemented. Therefore, we reasoned that one and half years after rolling out the TPT guide, health workers would have mastered the guide and the TPT program implementation would running effectively.

## Treatment procedure

The Uganda Ministry of Health recommended provision of TPT since 2006 under the national policy guidelines for TB/HIV collaborative activities [13] but supply chain interruptions required that PLHIV would be initiated on TPT based on available stock; a full six-month course had to be available for each patient initiation and once all stock was committed, new initiations would be halted. During the 100 days campaign significant stock of TPT was received and pushed to the facilities by National Medical Stores to initiate all eligible PLHIV.

Prior to initiation of TPT, PLHIV were first screened for signs and symptoms of TB using the TB case finding form and any presumptive case would be first investigated to rule out TB before TPT initiation. Those without TB symptoms would then be screened further to rule out

any contraindication to the initiation of TPT (see S2 Table X2 in S2 File, the checklists for screening for contraindications). Finally, an assessment of readiness to start TPT was performed using the 5A technique: Assess, Advise, Agree, Assist, and Arrange. Thereafter, a patients could be initiated on TPT along with pyridoxine according to the TPT guidelines [14]. As per the guidelines, PLHIV initiated on TPT should be reviewed two weeks after initiation to assess for severe side effects and reinforce adherence, and then received monthly refills and follow-up until they had completed 6 months of treatment. However, in programmatic setting, TPT refill visits are aligned with ART refill visits which can either be a month, 3 months, or 6 months, for better adherence through reduced missed visits, and save transport costs and time. TPT was stopped if a patient was diagnosed with TB or had severe side effects: jaundice, fever, severe tingling and burning sensation, blurred vision, loss of vision, convulsions, or unusual bleeding. Information on TPT was recorded in the paper-based registers at each health facility, whereas HIV care data were recorded in an electronic medical records (EMR) database system. During the study period, ART was provided according to the 2016 Uganda consolidated guidelines for prevention and treatment of HIV in Uganda [24].

## Evaluation of TPT uptake and outcomes

TPT initiation and treatment outcome information was extracted from paper-based TPT registers at each of the six healthy facilities studied. Because data was retrospectively collected, we additionally extracted prescribed drugs recorded in EMR to identify more individuals for whom TPT was prescribed during the study period.

## Data extraction and collection

The following HIV care data variables were extracted from the Uganda EMR database: Patients' socio-demographics (age, birthdates, gender, marital status), clinical factors (clinic visit date, weight in kilograms, height in meters, ART status at TPT initiation, ART start date, WHO stage, HIV/RNA viral load (absolute), pregnancy status, TB treatment history, Isoniazid prophylaxis start date, and place of residence). From the TPT registers the following data was extracted: IPT initiation date, TPT side effects, IPT outcome (completed 6-month TPT dose, loss to follow-up, died, stopped, transferred to another health facility), medical reasons for stopping TPT (including side effects, developed active TB).

A structured questionnaire for extraction of TPT information from the paper-based TPT register was designed and programmed into Open Data Kit (ODK) [25], a mobile data collection application. Data quality was controlled by applying data validation checks in the ODK database system, training of research assistants who extracted data from TPT registers and continuous review of data captured into ODK. TPT data was linked to HIV care data using the patient HIV clinic identifier number (i.e. IDCNO). If the identifier was missing in the TPT paper-based register, then patient characteristics including; names, age, sex, home address, ART start date were used if available. To maintain anonymity of patients, papers that were used to record patients identifying information for linking data were destroyed immediately before leaving the health facility on day of data collection. For the HIV care information, a list of variables was created to guide the process of data extraction from the Uganda EMR database version 3.2.0 at each of the six health facilities studied. A detailed list of information collected is presented in the data collection tools (See S3 File). Data were collected from July 2020 to March 2021. Permission to access patients' medical records in clinics studied was sought from the Directorate of Public Health and Environment, KCCA, Uganda.

## Statistical analysis

Data were analyzed using STATA software version 16.1 Special Edition (Stata Corp, College Station, Texas, USA).

We analyzed TPT uptake and completion in each quarter of the 4 years (2016–2019) to establish the trends. We examined change in TPT trends after the 100-days TPT accelerated campaigns in 2019. Baseline patient characteristics (listed in data collection tool attached as a S3 File) were defined at beginning of each year for the 4 years analyzed. For the analysis of TPT completion, baseline patient characteristics were defined as those at TPT initiation.

We used descriptive statistics (frequencies and percentages) to describe study participants across the four years studied. The TPT uptake was estimated as the proportion of patients newly initiated on TPT among PLHIV in HIV care who were eligible for TPT in a specific quarter and year. TPT completion was estimated as proportion of patients completed TPT among those initiated on TPT ≥6 months prior the campaign. Patients who were either lost-to-follow-up, died or stopped TPT were classified as non TPT completers whereas those who were transferred-out or had outcome not evaluated, were considered as missing outcomes. Analyses were performed on individual-level data, and TPT uptake and completion were fitted in regression models as binary outcomes (as Yes or No). TPT uptake analysis included all participants who were eligible for TPT and had information in the HIV care electronic database (EMR) whereas TPT completion analysis included only the participants whose IDCNOs matched in both the EMR database and the TPT paper-based registers. Also, since data extraction was truncated at 31st December 2019, only individuals initiated on TPT between 2016 quarter one to 2019 quarter two, were included in the TPT completion analysis to allow completion among individuals initiated on the 6-months TPT dose of isoniazid monotherapy. Factors associated with TPT uptake were examined by fitting Poisson regression with robust standard errors using generalized estimating equations (GEE) with exchangeable correlation structure to account for clustering of participants in the 6 clinics and repeated observation of participants over the 4 years [26]. When the prevalence of the outcome is >10%, the odds ratio exaggerates the relative risk or the prevalence ratio (PR) [27].

Factors associated with TPT completion were examined by fitting a Poisson regression model with cluster-correlated robust estimates of standard errors because it estimates the adjusted relative risks appropriately when the outcome is common (>10%, as for this analysis) than the logistic regression [28]. Also, Poisson regression with robust variance estimates was preferred over the log-binomial because the latter often face convergence problem which require proper choice of starting values [28]. A separate multivariable modified Poisson regression model was fitted for females to examine the association between pregnancy status and TPT completion. All multivariable analyses were adjusted for the predefined baseline factors (calendar quarter and year, gender, age groups, ART status, WHO stage, pregnancy status at TPT initiation). Statistical significance testing was based on a 2-tailed Wald test and 5% significance level.

To examine difference in TPT uptake and completion across sub-groups of predefined baseline characteristics listed above, stratified regression models were fitted by entering interactions between the variable for calendar quarter and year and each of the predefined baseline covariates, in separate adjusted models. Adjusted probabilities for TPT uptake and completion by the stratifying factors were then predicted from the fitted modified Poisson interaction models, using postestimation methods and results presented on line graphs.

We used variance inflation factors (VIFs) to evaluate multicollinearity in fitted models [29], and effect of missing data was assessed using the multiple imputation chained-equation (MICE) approach [30], worst-case and best-case scenarios [31].

## Ethics consideration

The AIDS Support Organization Research Ethics Committee (TASOREC), Kampala, Uganda (number: TASOREC/085/19-UG-REC-009) and the Uganda National Council for Science and Technology, Kampala, Uganda (UNCST number: HS729ES) granted ethical approval for the study. Due to the retrospective study design and that this was a public health surveillance, the need for patient consent was waived.

## Results

### Description of participants

Fig 1 shows the total numbers of patients who reported for HIV care at the six clinics by year: about 90% of patients were eligible for TPT during each year and were included in the analysis of TPT uptake. About 70% of patients initiated on TPT on overall, their data from the paper-based TPT registers could not be matched with their data in the HIV care electronic database. Analysis of IPT completion was performed on 10,131 patients for whom data in both paper-based TPT registers and the HIV care clinic electronic database were matched. For the patients evaluated and included in the analysis, data were missing on 10% and 4% patients for TPT uptake and treatment completion analyses, respectively. Across the 4 years, more than 70% were females (range: 73.5% -74.6%) and median age ranged from 33 to 34 year over the period of 4 years studied. About 15% of the females were pregnant (range: 14.6–16.0%). Almost all PLHIV were on ART due to test-and-treat expanded policy that was rolled out in 2016 in Uganda. However, because of the opt-out [24] option and that test and treat was not yet fully implemented in early 2016, some patients were not on ART; the proportion not on ART

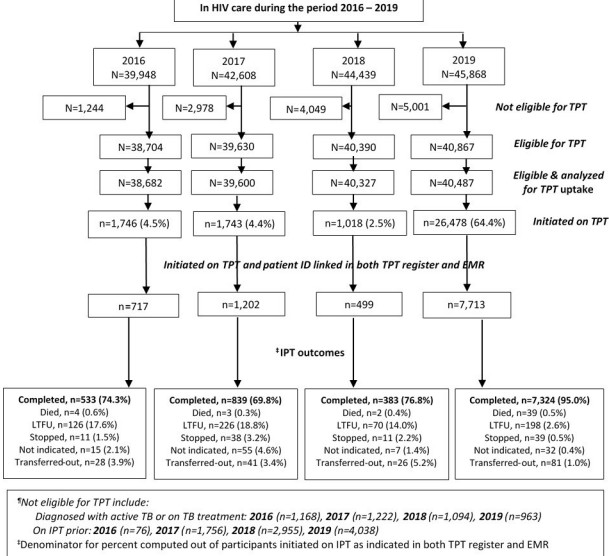

**Fig 1. Patients' flow diagram by the 4 years studied at the 6 health facilities.** 1) Eligible & analyzed for TPT uptake excluded the PLHIV eligible documented in the paper-based TPT register but had no information in the EMR. These were excluded because participants' characteristics were extracted from the EMR. Numbers excluded were: 22, 30, 63, 380 in 2016, 2017, 2018 and 2019 respectively. 2) Numbers initiated on TPT among total patients eligible for TPT by year and quarter: 2016Q1 (875/31,306), 2016Q2 (90/3,127), 2016Q3 (24/2,378), 2016Q4 (407/1,871), 2017Q1 (343/32,333), 2017Q2 (294/3,308), 2017Q3 (442/2,325), 2017Q4 (266/1,634), 2018Q1 (78/30,867), 2018Q2 (78/5,394), 2018Q3 (131/2,251), 2018Q4 (296/1,815), 2019Q1 (4,802/25,694), 2019Q2 (2,448/11,061), 2019Q3 (1,269/2,073), 2019Q4 (516/1,658). 3) Majority of patients were scheduled to return to the HIV clinic for review every after 3 months. Number of patients visited during the quarter don't add up to the unique numbers visited during the year.

declined overtime from 9% in 2016 to about 2% in 2019. All collected baseline characteristics are shown in Table 1.

## Trends of TPT uptake

Among the eligible patients at the selected health facilities, TPT uptake was low below 25% (ranged between 0.3% and 22%) during the period from 2016 quarter one to 2019 quarter two, with a sharp decline at 2017 quarter one and 2018 quarter one. It however increased by about 3-folds from 22.1% to 61.2%in 2019 quarter two (i.e. before the roll-out of the 100-day accelerated TPT intervention) and quarter three (i.e. after the roll-out of the 100-day accelerated TPT intervention) respectively. There was a drop of TPT uptake to 31% in 2019 quarter four, nonetheless, it was still above that observed during the earlier period before the 100-days accelerated TPT campaign (Fig 2A). At multivariable analysis using modified Poisson GEE model, TPT uptake was 2.6 times higher in 2019 quarter three (i.e. after roll-out of the 100-days TPT

**Table 1. Patients' baseline characteristics at TPT uptake by year.**

| Characteristics[†] | 2016 | 2017 | 2018 | 2019 |
|---|---|---|---|---|
| Total patients who had at least one clinic encounter during the year and were eligible for TPT, n (%) \|\| | 38,682 | 39,600 | 40,327 | 40,487 |
| **Demographic factors** | | | | |
| Sex | | | | |
| Male | 9,804 (25.4) | 10,075 (25.4) | 10,548 (26.2) | 10,731 (26.5) |
| Female | 28,878 (74.6) | 29,525 (74.6) | 29,779 (73.8) | 29,756 (73.5) |
| Age in years | | | | |
| Median (inter-quartile range) | 33 (27, 40) | 33 (28, 41) | 34 (28, 41) | 34 (28, 42) |
| Age categories, n (%) | | | | |
| 15–19 | 855 (2.2) | 782 (2.0) | 744 (1.8) | 687 (1.7) |
| 20–24 | 4,063 (10.5) | 3,775 (9.5) | 3,552 (8.8) | 3,262 (8.1) |
| 25–34 | 15,959 (41.3) | 16,046 (40.5) | 15,994 (40.0) | 15,484 (38.2) |
| 35+ | 17,805 (46.0) | 18,997 (48.0) | 20,037 (49.7) | 21,054 (52.0) |
| **ART information** | | | | |
| On ART, n (%) | 35,289 (91.2) | 37,672 (95.1) | 39,375 (97.6) | 39,737 (98.2) |
| **Clinical factors** | | | | |
| WHO stage, n (%) [§] | | | | |
| 1 or 2 | 34,279 (89.3) | 36,348 (92.5) | 38,448 (96.0) | 39,011 (97.1) |
| 3 or 4 | 4,106 (10.7) | 2,946 (7.5) | 1,594 (4.0) | 1,154 (2.9) |
| BMI kg/m$^2$, n (%)[§] | | | | |
| <18.5 | 3,816 (11.5) | 3,861 (10.8) | 3,664 (10.1) | 3,455 (9.2) |
| ≥18.5 | 29,242 (88.5) | 31,917 (89.2) | 32,585 (89.9) | 34,073 (90.8) |
| HIV Viral load copies/mL, n (%) | | | | |
| <1000 | 8,824 (91.3) | 6,863 (82.8) | 13,639 (91.3) | 21,828 (95.2) |
| ≥1000 | 840 (8.7) | 1,421 (17.2) | 1,299 (8.7) | 1,104 (4.8) |
| **Other conditions** | | | | |
| Pregnant[¶] | 4,638 (16.0) | 4,606 (15.6) | 4,335 (14.6) | 4,613 (15.5) |

[†]Characteristics of patients eligible for TPT at their first visit at the clinic in a specific year.

\|\| Included only participants who appeared in the EMR since it was the source document with detailed participants' characteristics. Number of eligible participants presented here compared to Fig 1, are less by number of participants who were in paper-based TB registers but not in the EMR, i.e. 22, 30, 63, 380 in 2016, 2017, 2018 and 2019 respectively.

§ Missing values: WHO stage (0.8%), BMI (10.4%)

¶ Denominator is number of females included in analysis

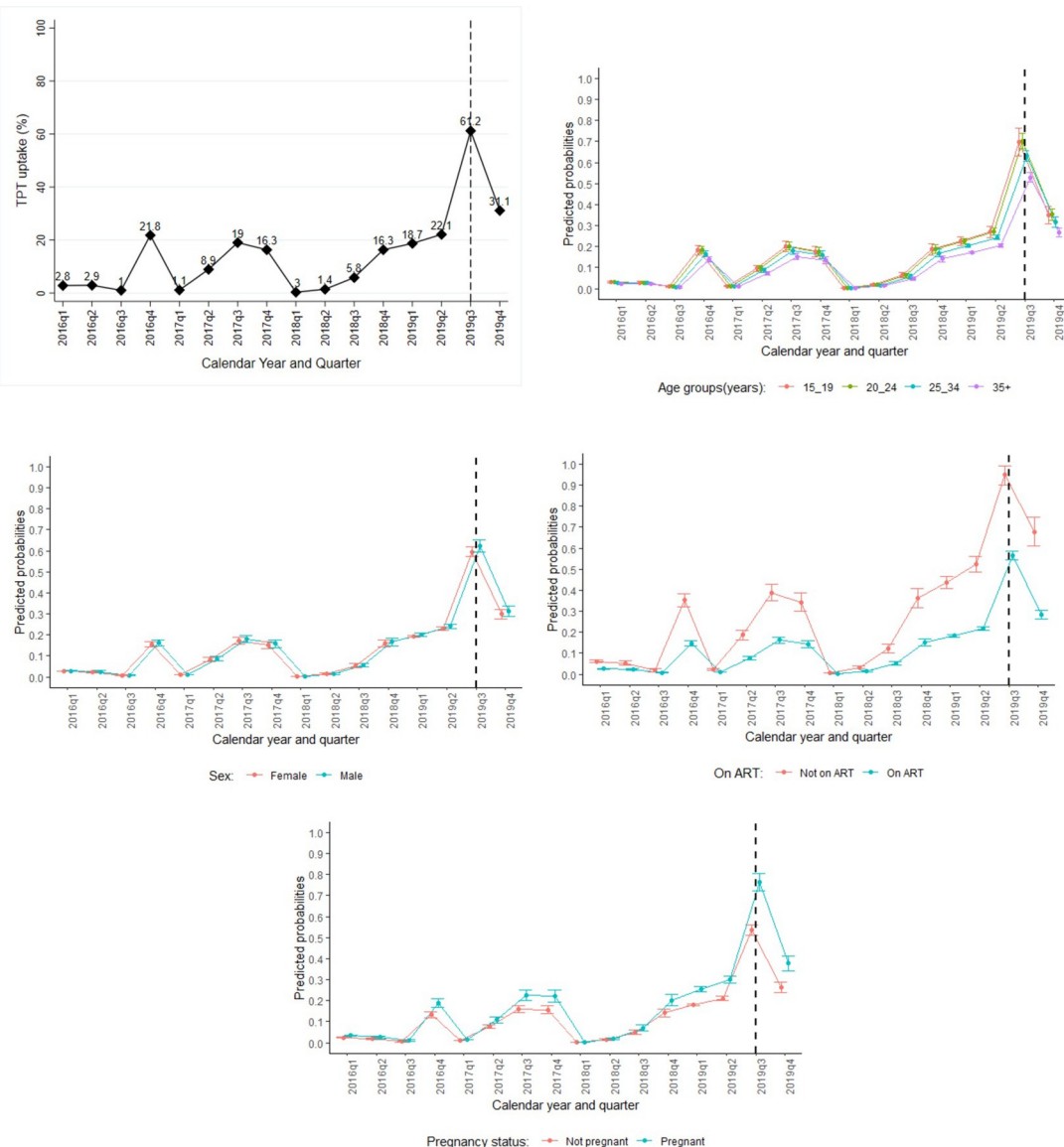

**Fig 2. a: Trend of TPT uptake across calendar years and quarters.** q1, q2, q3, and q4 denote quarters 1,2,3, and 4 respectively. The vertical black dashed line at 2019q3 indicates a point when the 100-day TPT accelerated campaign was rolled-out. **b: Trend of predicted probabilities of TPT uptake by age groups.** q1, q2, q3, and q4 denote quarters 1,2,3, and 4 respectively. The vertical black dashed line at 2019q3 indicates a point when the 100-day TPT accelerated campaign was rolled-out. Points on line graph represent predicted probabilities of TPT uptake from GEE modified Poisson regression model with cluster standard errors to account for correlation of outcomes within each of the 6 clinics, adjusting for sex, age groups, ART status, WHO stage. Model was fitted with interaction between quarter of calendar year and age groups covariates. **c: Trend of predicted probabilities of TPT uptake by sex.** q1, q2, q3, and q4 denote quarters 1,2,3, and 4 respectively. The vertical black dashed line at 2019q3 indicates a point when the 100-day TPT accelerated campaign was rolled-out. Points on line graph represent predicted probabilities of TPT uptake from modified Poisson GEE regression model with cluster standard errors to account for correlation of outcomes within each of the 6 clinics, adjusting for sex, age groups, ART status, WHO stage. Model was fitted with interaction between quarter of calendar year and sex covariates. **d: Trend of predicted probabilities of TPT uptake by ART status.** q1, q2, q3, and q4 denote quarters 1,2,3, and 4 respectively. The vertical black dotted at 2019q3 line indicates a point when the 100-day TPT accelerated campaign was rolled-out. Points on line graph represent predicted probabilities of TPT uptake from modified Poisson GEE regression model with cluster standard errors to account for correlation of outcomes within each of the 6 clinics, adjusting for sex, age groups, ART status, WHO stage. Model was fitted with interaction between quarter of calendar year and sex covariates. **e: Trend of predicted probabilities of TPT uptake by pregnancy status.** q1, q2, q3, and q4 denote quarters 1,2,3, and 4 respectively. The vertical black dotted at 2019q3 line indicates a point when the 100-day TPT accelerated campaign was rolled-out. Points on line graph represent predicted probabilities of TPT uptake from modified Poisson GEE regression model with cluster standard errors to account for correlation of outcomes within each of the 6 clinics, adjusting for sex, age groups, ART status, WHO stage. Model was fitted with interaction between quarter of calendar year and sex covariates.

intervention) compared to that in 2019 quarter two (i.e. before the 100-days TPT intervention roll-out) (adjusted Prevalence Ratio [aPR] = 2.58, 95% confidence interval [95%CI] = 2.45, 2.72, and P value<0.001). The analysis was adjusted for baseline characteristics at each patient's first visit in each year (sex, age categories, ART status, and WHO stage) (Table 2). TPT uptake did not significantly vary across clinics overtime (S2 Table 3a in S2 File).

Analysis of TPT uptake across selected participants' characteristics indicated: trends of TPT uptake were similar across age groups and sex of the participants over the 4 years studied (Fig 2B, 2C and 2E). Before 2018, TPT uptake was significantly higher among patients who were not on ART compared to those who were on ART (predicted mean probabilities of TPT ranged from 0.6% to 90.0% among not on ART group and 0.3% to 56.4% among ART group) (Fig 2D). Also, TPT uptake was slightly higher among pregnant women compared to their non-pregnant counterparts (predicted probabilities ranged from 0.2% to 55.5% among non- pregnant women and 0.3% to 76.2% among pregnant women) (Fig 3B). Nonetheless, starting from 2019 quarter three (i.e. after roll-out of the 100-day TPT accelerated campaign), TPT uptake increased uniformly in all categories.

## Trends of TPT completion

In the subset of patients in whom TPT completion was evaluated, it was consistently high at above 70% at most of the time even in patients initiated on TPT during the period studied. However, there was a drop in TPT completion rates during 2017 quarter one and 2018 quarter three. TPT completion rate systematically increased among PLHIV who were initiated during the period from 2018 quarter four onwards (i.e. the period targeted by the 100-day TPT intervention), it increased from 71.9% for those initiated during 2018 quarter three to 83.2%, 95.3% and 97.1% among those initiated during 2018 quarter 4 and 2019 quarters 1, and 2, respectively (Table 3 and Fig 3A). The increase in TPT completion during this period remained significant even after adjusting for patients' baseline characteristics (aPR [95%CI] = 1.09 [1.04, 1.14] and P value<0.001, 1.10 [1.05,1.15] and P value<0.001 among individuals initiated during 2019 quarter 1, and 2, respectively compared to those initiated in 2018 quarter 4). TPT completion was significantly higher among patients aged 25 years and above compared to those below 25 years (aPR [95%CI] = 1.07 [0.98, 1.16], 1.11 [1.01, 1.23], 1.14 [1.02, 1.26] for ages 20–24, 25–34 and 35+years respectively with 15–19 years as reference age group). Patients not on ART or newly initiated on ART at TPT start were significantly associated with lower TPT completion compared to the ART experienced (aPR [95%CI] = 0.84 [0.73, 0.97], 0.94 [0.90, 0.97], respectively). There was no sufficient evidence for the association of TPT completion with sex (Wald P value = 0.106) and WHO stage (Wald P value = 0.161). There was a borderline significance of TPT completion by pregnancy status, where pregnant females were associated with lower completion rate compared to their non-pregnant female counterparts (aPR [95%CI] = 0.96 [0.93, 0.99] and P value = 0.043) (Table 3).

In a sensitivity analyses, there was no substantial change in estimates (both prevalence ratios and 95%CI) of TPT completion comparing the first quarter targeted for TPT completion by the 100-day accelerated TPT campaign (i.e. 2018 quarter 4, used as reference) and subsequent quarters (i.e. 2019 quarters 1,2,and 3), after refitting the model using sensitivity analyses on missing outcomes: worst-case scenario (aPR [95%CI] = 1.09 [1.04, 1.14] and P value = 0.001, 1.10 [1.04,1.16] and P value<0.001, for 2019 quarters 1 and 2, respectively), best-case scenario (aPR [95%CI] = 1.08 [1.03, 1.13] and P value = 0.001, and 1.09 [1.04,1.14] and P value<0.001, for 2019 quarters 1 and 2, respectively) and multiple imputation (aPR [95%CI] = 1.08 [1.03, 1.14] and P value = 0.002, and 1.09 [1.04,1.15] and P value = 0.001 for 2019 quarters 1 and 2, respectively) (S2 Table 4 in S2 File).

**Table 2. Factors associated with TPT uptake.**

| Factors | Crude PR (95%CI) † | P-value | Adjusted PR (95%CI) † | P-value |
|---|---|---|---|---|
| **Calendar year and quarter**\* | | | | |
| 2016 Q1 | 0.13 (0.12,0.14) | <0.001 | 0.12 (0.11,0.13) | <0.001 |
| 2016 Q2 | 0.13 (0.11,0.16) | <0.001 | 0.10 (0.08,0.13) | <0.001 |
| 2016 Q3 | 0.05 (0.03,0.07) | <0.001 | 0.04 (0.02,0.05) | <0.001 |
| 2016 Q4 | 0.99 (0.90, 1.09) | <0.001 | 0.68 (0.62,74) | <0.001 |
| 2017 Q1 | 0.05 (0.04,0.05) | <0.001 | 0.05 (0.04,0.05) | <0.001 |
| 2017 Q2 | 0.40 (0.36,0.45) | <0.001 | 0.36 (0.32,0.40) | <0.001 |
| 2017 Q3 | 0.84 (0.76,0.92) | <0.001 | 0.74 (0.68,0.82) | <0.001 |
| 2017 Q4 | 0.73 (0.65,0.82) | <0.001 | 0.65 (0.58,0.74) | <0.001 |
| 2018 Q1 | 0.01 (0.01,0.02) | <0.001 | 0.01 (0.01,0.02) | <0.001 |
| 2018 Q2 | 0.07 (0.05,0.08) | <0.001 | 0.06 (0.05,0.08) | <0.001 |
| 2018 Q3 | 0.25 (0.21,0.30) | <0.001 | 0.23 (0.20,0.28) | <0.001 |
| 2018 Q4 | 0.75 (0.67,0.83) | <0.001 | 0.69 (0.62,0.77) | <0.001 |
| 2019 Q1 | 0.85 (0.81,0.88) | <0.001 | 0.84 (0.80,0.87) | <0.001 |
| 2019 Q2¶ | 1 | | 1 | |
| 2019 Q3 | 2.78 (2.65,2.92) | <0.001 | 2.58 (2.45,2.72) | <0.001 |
| 2019 Q4 | 1.40 (1.29,1.52) | <0.001 | 1.30 (1.19,1.41) | <0.001 |
| **Demographic factors** | | | | |
| Sex | | | | |
| Male | 1 | | 1 | |
| Female | 0.94 (0.90, 0.97) | 0.001 | 0.96 (0.92, 0.99) | 0.017 |
| Age in years, n (%) | | | | |
| 15–19 | 1 | | 1 | |
| 20–24 | 0.92 (0.82, 1.02) | 0.121 | 1.00 (0.91, 1.11) | 0.954 |
| 25–34 | 0.72 (0.65, 0.80) | <0.001 | 0.90 (0.82, 0.99) | 0.036 |
| 35+ | 0.59 (0.53, 0.65) | <0.001 | 0.76 (0.69, 0.83) | <0.001 |
| **Clinical factors** | | | | |
| On ART | | | | |
| No | 1 | | 1 | |
| Yes | 0.54 (0.51, 0.58) | <0.001 | 0.42 (0.39, 0.45) | <0.001 |
| WHO stage | | | | |
| 1 or 2 | 1 | | 1 | |
| 3 or 4 | 0.63 (0.57, 0.69) | <0.001 | 0.80 (0.73, 0.88) | <0.001 |

\* Number of completed case analyzed (Year2016 n = 33,021, Year2017 n = 35,718, Year2018 n = 36,198, Year2019 n = 37,418).

Q denotes quarter of calendar year.

Proportion of missing data on at-least one of the variables per year in adjusted model, (Year2016 [15%], Year2017 [10%], Year2018 [10%], Year2019 [8%]).

† **PR**–Prevalence Ratio estimated using Generalized Estimating Equations (GEE) with modified Poisson regression model and robust standard errors accounting for repeated observation of participants over the 4 years

CI–confidence interval

¶ 2019 Q2 was chosen as reference on the calendar year and quarter covariate to allow compare the effect before and after of the 100-day TPT accelerated campaign rolled-out at beginning of 2019 Q3 (i.e. July 2019).

†Refitting the same model for women only to examine TPT uptake by pregnancy status, adjusted Prevalence Ratio (Apr) for TPT uptake, aPR = 1.42 (95%CI = 1.36, 1.49), P value <0.001, among pregnant compared non-pregnant (reference category)

Looking at TPT completion trends by selected participants' characteristics: Before 2018 quarter four (i.e. the period outside window targeted by the 100-days TPT intervention), on

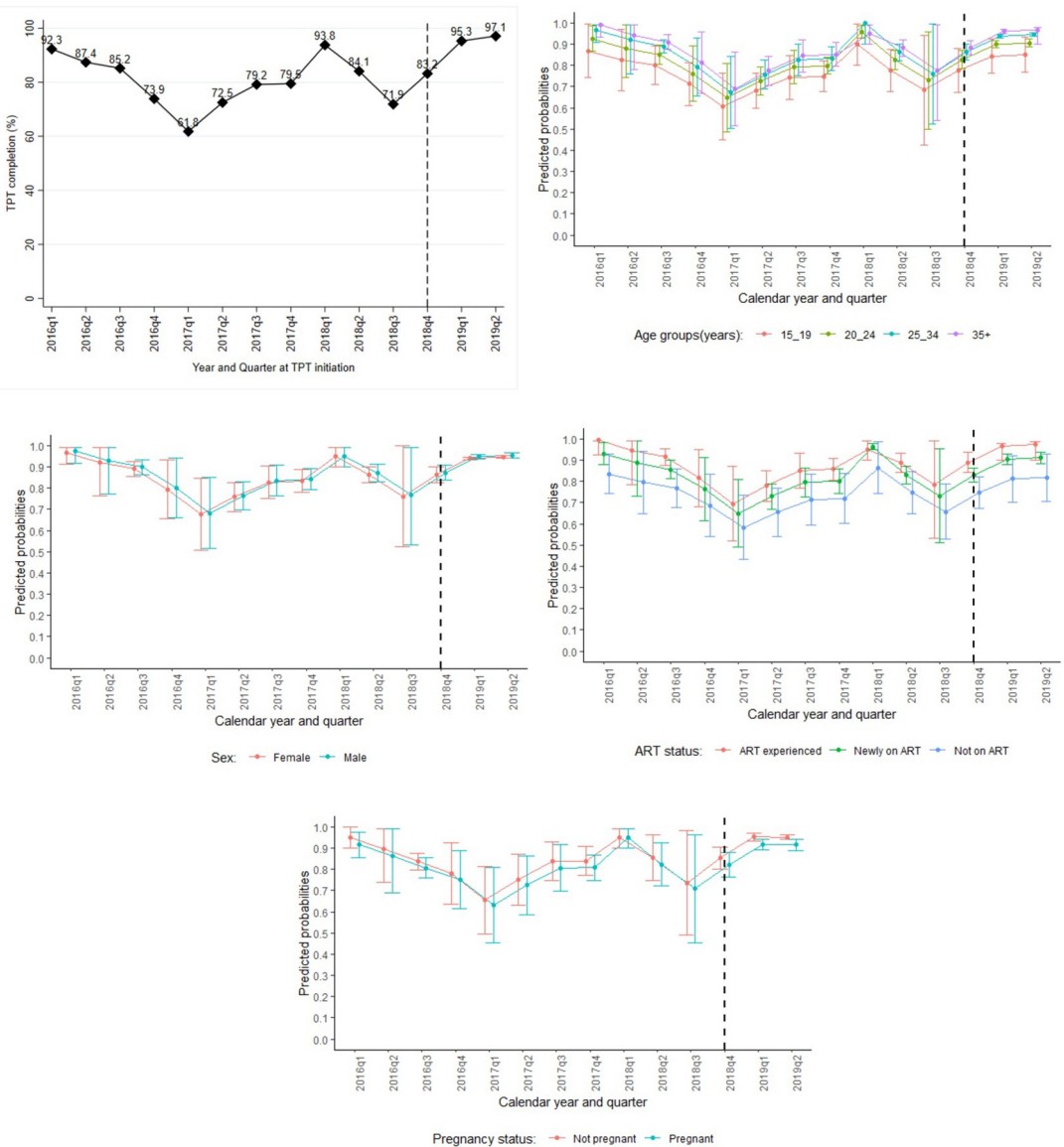

**Fig 3. a: Trend of TPT completion by calendar years and quarters.** q1, q2, q3, and q4 denote quarters 1,2,3, and 4 respectively. The vertical dotted line at 2018q4 marks a point from which the 100-day TPT accelerated intervention targeted to achieve 100% completion. Trend graph truncated at 2019q2, a point which PLHIV initiated on TPT during the period studied were expected to have completed their 6-months TPT dose. **b: Trend of predicted probabilities of TPT completion by age groups.** q1, q2, q3, and q4 denote quarters 1,2,3, and 4 respectively. The vertical dotted line at 2018q4 marks a point from which the 100-day TPT accelerated intervention targeted to achieve 100% completion. Trend graph truncated at 2019q2, a point which PLHIV initiated on TPT during the period studied were expected to have completed their 6-months TPT dose. **c: Trend predicted probabilities of TPT uptake by sex.** q1, q2, q3, and q4 denote quarters 1,2,3, and 4 respectively. The vertical dotted line at 2018q4 marks a point from which the 100-day TPT accelerated intervention targeted to achieve 100% completion. Trend graph truncated at 2019q2, a point which PLHIV initiated on TPT during the period studied were expected to have completed their 6-months TPT dose. **d: Trend of TPT completion by ART status.** q1, q2, q3, and q4 denote quarters 1,2,3, and 4 respectively. The vertical dotted line at 2018q4 marks a point from which the 100-day TPT accelerated intervention targeted to achieve 100% completion. Trend graph truncated at 2019q2, a point which PLHIV initiated on TPT during the period studied were expected to have completed their 6-months TPT dose. **e: Trend predicted probabilities of TPT completion by pregnancy status.** q1, q2, q3, and q4 denote quarters 1,2,3, and 4 respectively. The vertical dotted line at 2018q4 marks a point from which the 100-day TPT accelerated intervention targeted to achieve 100% completion. Trend graph truncated at 2019q2, a point which PLHIV initiated on TPT during the period studied were expected to have completed their 6-months TPT dose.

**Table 3. TPT completion and associated factors.**

| Factor | Total* | TPT completion, n (%) | Crude PR (95%CI) [†] | P-value | Adjusted PR (95%CI) [†] | P-value |
|---|---|---|---|---|---|---|
| Calendar year and quarter | | | | | | |
| 2016 Q1 | 65 | 60 (92.3) | 1.11 (1.04,1.18) | 0.001 | 1.12 (1.06,1.18) | <0.001 |
| 2016 Q2 | 103 | 90 (87.4) | 1.05 (0.91,1.21) | 0.509 | 1.07 (0.90,1.27) | 0.479 |
| 2016 Q3 | 81 | 69 (85.2) | 1.02 (0.97,1.08) | 0.443 | 1.03 (0.99,1.07) | 0.164 |
| 2016 Q4 | 425 | 314 (73.9) | 0.88 (0.72,1.09) | 0.244 | 0.92 (0.76,1.11) | 0.389 |
| 2017 Q1 | 123 | 76 (61.8) | 0.75 (0.58,0.98) | 0.037 | 0.78 (0.61,1.01) | 0.055 |
| 2017 Q2 | 247 | 179 (72.5) | 0.87 (0.77,0.98) | 0.024 | 0.88 (0.79,0.98) | 0.017 |
| 2017 Q3 | 395 | 313 (79.2) | 0.95 (0.86,1.05) | 0.325 | 0.96 (0.87,1.05) | 0.377 |
| 2017 Q4 | 341 | 271 (79.5) | 0.96 (0.88,1.04) | 0.304 | 0.96 (0.89,1.05) | 0.394 |
| 2018 Q1 | 16 | 15 (93.7) | 1.13 (1.06,1.20) | <0.001 | 1.16 (1.11,1.22) | <0.001 |
| 2018 Q2 | 88 | 74 (84.1) | 1.01 (0.96,1.06) | 0.709 | 0.99 (0.95,1.05) | 0.961 |
| 2018 Q3 | 64 | 46 (71.9) | 0.85 (0.65,1.13) | 0.265 | 0.88 (0.67,1.15) | 0.356 |
| 2018 Q4¶ | 298 | 248 (83.2) | 1 | | 1 | |
| 2019 Q1 | 3,476 | 3,314 (95.3) | 1.15 (1.09,1.20) | <0.001 | 1.09 (1.04,1.14) | <0.001 |
| 2019 Q2 | 3,889 | 3,777 (97.1) | 1.17 (1.11,1.23) | <0.001 | 1.10 (1.05,1.15) | <0.001 |
| **Demographic factors** | | | | | | |
| Sex | | | | | | |
| Male | 2,653 | 2,475 (93.3) | 1 | | 1 | |
| Female | 6,869 | 6,298 (91.7) | 0.98 (0.97, 0.99) | 0.014 | 0.99 (0.98, 1.00) | 0.106 |
| Age in years, n (%) | | | | | | |
| 15–19 | 203 | 162 (79.8) | 1 | | 1 | |
| 20–24 | 948 | 789 (83.2) | 1.04 (0.91, 1.19) | 0.513 | 1.07 (0.98, 1.16) | 0.142 |
| 25–34 | 3,880 | 3,517 (90.6) | 1.14 (0.99, 1.30) | 0.068 | 1.11 (1.01, 1.23) | 0.038 |
| 35+ | 4,491 | 4,305 (95.9) | 1.20 (1.06, 1.36) | 0.004 | 1.14 (1.02, 1.26) | 0.016 |
| **Clinical factors** | | | | | | |
| ART status | | | | | | |
| Not on ART | 331 | 239 (72.2) | 0.74 (0.61, 0.91) | 0.004 | 0.84 (0.73, 0.97) | 0.017 |
| Newly on ART[‡] | 2,700 | 2,230 (82.6) | 0.85 (0.80, 0.90) | <0.001 | 0.94 (0.90, 0.97) | 0.001 |
| ART experienced | 6,491 | 6,304 (97.1) | 1 | | 1 | |
| WHO stage[§] | | | | | | |
| 1 or 2 | 9,254 | 8,538 (92.3) | 1 | | 1 | |
| 3 or 4 | 268 | 235 (87.7) | 0.95 (0.93, 0.97) | <0.001 | 0.97 (0.94, 1.01) | 0.161 |

* Number of completed case analyzed, N = 9,522/9,895 (96%). Missing on at-least one of the variables in adjusted model, n = 373/10131 (4%).

[†] **PR**–Prevalence Ratio estimated using modified Poisson regression model with cluster standard errors to account for clustering since data was collected from different clinics.

CI–confidence interval

[§] Missing values: outcome (n = 285, 3%), WHO stage (n = 98, 1%),

[‡] Newly on ART includes participants who were on ART for ≤ 3 months at TPT initiation.

Refitting the same model for women only to examine TPT completion by pregnancy status, adjusted Prevalence Ratio for TPT completion = 0.96 (95%CI = 0.93, 0.99), P value = 0.043, among pregnant compared non-pregnant (reference category)

overall TPT completion was lower among young PLHIV aged 24 years and below (i.e. predicted mean probabilities of TPT completion ranged from 60.7% to 95.8%) compared to the adults aged 25 year and above (i.e. predicted mean probabilities of TPT completion ranged from 67.5% to 99.9%). Though this was not statistically significant because the confidence intervals were overlapping (Fig 3B). TPT completion was always lower among those not on

ART (i.e. predicted mean probabilities of TPT completion ranged from 58.2% to 86.2%), and among newly initiated on ART (i.e. predicted mean probabilities of TPT completion ranged from 65.0% to 96.3%) compared to the ART experienced (i.e. predicted mean probabilities of TPT completion ranged from 69.5% to 99.9%), though the confidence intervals were overlapping (Fig 3D). Also, TPT completion was slightly lower among pregnant females (i.e. predicted mean probabilities of TPT completion ranged from 63.1% to 99.0%) compared to non-pregnant females (i.e. predicted mean probabilities of TPT completion ranged from 65.61% to 99.9%), though the confidence intervals were overlapping (Fig 3E). Nonetheless, TPT completion uniformly increased across all groups during the period targeted by the 100-days TPT intervention (i.e. from 2018 quarter four to 2019 quarter two).

For the TPT non-completers, the proportions of those who died were consistently below 1% (ranged: 0% to 1.0%), whereas proportions of LTFUs were a bit high at an average of 16% before 2019 (ranged from 0% to 34%) but dropped systematically in 2019 (106/3541 [3.0%], 90/3936 [2.3%], 2/236 [0.9%] in quarters 1,2 and 3 respectively). LTFU was more pronounced among participants aged <25years, not on ART or newly on ART and pregnant females compared to their counterparts (S2 Table 1 in S2 File).

## Discussion

The 100-day campaign was the major intervention undertaken by the Ugandan Ministry of Health in 2019 and it increased TPT uptake overall. Despite the large number of TPT initiation, the completion rates also increased substantially. This was achieved using existing health care resources, system strengthening, multi-stakeholder engagement in the campaign, and enhanced TPT delivery, monitoring and reporting [18]. The campaign objective was to enroll 300,000 PLHIV on TPT in 100 days in order to improve TPT coverage from about 30% before the campaign (July 2019) to at least 50% of all the PHIV in care by the end of the campaign (October 2019) and ensure 100% TPT completion by 30th September 2019 for PLHIV enrolled on TPT during the period: October–December 2018, and January–February 2019 before the campaign. The drops in TPT uptake observed in 2017 quarter one and 2018 quarter one was due to general stockouts at the National Medical Stores and health facilities were affected. Whereas the decline in TPT uptake in 2019 quarter four could be because the 100-day campaign had ended and most of the eligible PLHIV who were active in care had been initiated on TPT in these clinics. Based on these results, we can conclude that the 100-day TPT campaign in Uganda was highly successful. Given that many resource-limited settings are still struggling to achieve high rates of TPT coverage among the PLHIV, this is an intervention that we can recommend in such settings to improve TPT program.

Such interventions can help to both create awareness and improve performance of national programs that are static. A massive rollout of medication, however, may exacerbates occurrence of side effects in the participants involved, and this should be adequately monitored and documented through pharmacovigilance. Also, for a successful program there should be a good organizational structure and sufficient training of frontline health workers to appreciate and ably implement the intervention. These, however, require logistics and human resources which may not be sustainable in a long run especially in resource-constrained countries, in turn this may undo the immediate benefits achieved in a short run.

TPT uptake was higher among newly enrolled PLHIV in care not on ART and among pregnant women living with HIV before 2018. This was due to unstable TPT supplies in the country before 2018, and the national guidance to health facilities providing care to PLHIV was to prioritize TPT initiation to the groups that were most vulnerable of developing TB, e.g. the newly enrolled PLHIV in care not on ART; however, in 2019, there was an increased TPT

supplies at the national level which improved TPT enrollment to all PLHIV who had never previously received TPT [18]. With the increased TPT supplies at national level and increased TPT availability during the 100-day campaign in 2019 [18], we observed improved uptake across all patient groups. Previously, low TPT uptake had been reported in another prospective study conducted in 2017 in Uganda by Kalema et al. [16] where of 372 PLHIV participants who were eligible, only 17% initiated TPT. Although it was impossible to explore reasons for poor TPT uptake given the retrospective nature of our study, some of the reasons cited for low TPT uptake highlighted in Kalema's study included: limited capacity of clinicians to exclude TB using symptoms alone, fear of promoting drug resistance due to isoniazid monotherapy and inconsistent TPT drug supplies [16]. To address these barriers during the 100-day campaign implemented in Uganda: job aids and guidelines were provided to health workers implementing the program to improve their knowledge on TPT provision. Also, in order to ensure drug stock security and reduced stock-outs at a health facility, isoniazid and vitamin B6 stocks were mobilized from warehouses and unutilized stocks from some other health facilities, and used at health facilities with high demand. Additionally, there was capacity building of teams to manage and forecast future drug requirement at health facilities [18].

In prospective research studies, many countries in sub-Saharan Africa (SSA) have shown similar success with TPT completion rates above 80% [7, 32, 33]. In this report we show high completion rates of TPT in a program setting over the entire period studied and it further increased after the roll-out of the 100-days TPT campaign. The high TPT completion rate was achieved because PLHIV would be initiated on TPT by committing a full six months course of isoniazid monotherapy for each patient and once all stock is committed, there would be no new TPT initiation until new stock is available. This helped to ensure that those initiated complete their TPT course regardless of drug stock-outs. However, during the 100-days campaign, significant stock of TPT was received and pushed to the health facilities by NMS to initiate all eligible all PLHIV and enable completion of those already initiated. The high TPT completion rates during the 100-days TPT campaign was also attributed to: good coordination between the AIDS Control and National TB and Leprosy Programs, consistent supply of TPT and vitamin B6, substantial support from partners like PEPFAR, United States Agency for International Development (USAID), Centers for Disease Control (CDC), among other, partner engagement, facility mentorship activities and improved reporting [18].

We observed that patients not on ART or newly initiated on ART were associated with low IPT completion before 2019 compared to their ART experienced counterparts. Similarly, some previous studies found that TPT completion was associated with being on ART [7, 34]. Patients who are new or not on ART are likely to have stigma [35, 36], poor adherence [37], and lack of understanding of the role of TB prevention in the absence of symptoms [38].

The higher TPT completion among older persons (aged ≥25 years) compared to that among younger persons (aged 15–24 years) found in our study could be attributed to poor treatment adherence among young persons [37]. This calls for the better strategies focusing at improvement treatment adherence among young PLHIV in order to achieve better TPT completion rates.

## Limitations

In this retrospective review of routinely collected data, we had the following limitations: firstly, there is a possibility that we included some TPT ineligible individuals due to undocumented comorbidities or TB symptoms status. This could have exaggerated the denominator leading to under-estimation of TPT uptake. Secondly, for some patients, HIV clinic identifier numbers were missing or incorrectly recorded in the paper-based TPT registers and thus, TPT data for

those patients could not be linked to their HIV data. Patients whose TPT and HIV data could not be linked, were excluded from the TPT completion analysis which could have biased the TPT completion estimates. Like in any other retrospective analysis of routinely collected data, missing data is inevitable. However, we found no substantial difference in both estimates and 95% confidence intervals from complete case analysis and sensitivity analyses. Lastly, we could not report on TPT side effects because this information was largely missing in the data sources. Results from our study can be generalized to other similar resource-limited settings with high of TB and HIV coinfections.

## Conclusion

The 100-day TPT campaign intervention in Uganda was highly successful in initiation and completion of TPT among PLHIV and overcame many of the barriers that have plagued scale up of TPT in national programs. The combination of sensitization, training, drugs stocks, securing a full TPT dose for each PLHIV initiated on TPT, and on-site support and monitoring led to good results. We recommend more counselling sessions for pregnant women living with HIV, PLHIV below the age of 25 years and those newly enrolled into HIV care or newly on ART to ensure better TPT completion rates. As Uganda rolls out rifapentine and isoniazid once weekly for 3 months (3HP) as per the national TPT guidelines [15], young people living with HIV should be prioritized for this regimen given its shorter duration, better completion rates and fewer adverse events [5]. Importantly, there is need for consistent TPT supply at health facilities if gains we observed in this evaluation are to be sustained. Future evaluation with more years of data post 100-day accelerated TPT intervention is required to assess the sustainability of these gains.

## Supporting information

**S1 File.**
(XLSX)

**S2 File. Additional information and analysis.**
(DOCX)

**S3 File.**
(DOCX)

## Acknowledgments

The program is supported by the Presidents Emergency Plan for AIDS Relief (PEPFAR) through the United States Centers for Disease Control (CDC) and Prevention (terms of Cooperative Agreement NU2GGH002022).

The authors acknowledge contributions of various staff from the Infectious Diseases Institute: Ms. Aidah Nanvuma, Mr. Grace Banturaki, Mr. Godwin Anguzu, Mr. Edison Katunguka; Research assistants: Mr. Daniel Kirumira, Ms. Jessica Masika, Ms. Carol Kaidu, Ms. Patricia Asiimwe, Mr. Martin Atugonza, Ms. Namada Moureen Juma; staff at the KCCA Directorate of Public Health and Environment office in Uganda: Dr. Daniel Okello, Ms. Phiona Nazziwa; and staff and in-charges at the 6 KCCA health care facilities who assisted in retrieval of patients' records. We finally thank the CDC staff who reviewed the manuscript: Deus Lukoye, Grace Namayanja, Donna Kabatesi, Nelson Lisa J, and Ssempiira Julius.

## Author Contributions

**Conceptualization:** Joseph Musaazi.

**Data curation:** Joseph Musaazi.

**Formal analysis:** Joseph Musaazi.

**Funding acquisition:** Joseph Musaazi.

**Methodology:** Joseph Musaazi.

**Project administration:** Joseph Musaazi.

**Supervision:** Joseph Musaazi, Christine Sekaggya-Wiltshire, Agnes Kiragga.

**Validation:** Joseph Musaazi.

**Writing – original draft:** Joseph Musaazi.

**Writing – review & editing:** Joseph Musaazi, Christine Sekaggya-Wiltshire, Stephen Okoboi, Stella Zawedde-Muyanja, Mbazi Senkoro, Nelson Kalema, Paul Kavuma, Proscovia M. Namuwenge, Yukari C. Manabe, Barbara Castelnuovo, Agnes Kiragga.

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
