## [Editor Report · Decision Letter 0]

13 Jun 2022

PONE-D-22-13680Increased uptake of tuberculosis preventive therapy (TPT) among people living with HIV following the 100-days accelerated campaign: A retrospective review of routinely collected data at six urban public health facilities in UgandaPLOS ONE

Dear Dr. Musaazi,

Thank you for submitting your manuscript to PLOS ONE. After careful consideration, we feel that it has merit but does not fully meet PLOS ONE’s publication criteria as it currently stands. Therefore, we invite you to submit a revised version of the manuscript that addresses the points raised during the review process.

We look forward to receiving your revised manuscript.

Kind regards,

Esther Buregyeya, M.D

Academic Editor

PLOS ONE

Journal Requirements:

"Support for data collection was provided by EDCTP, grant number: 1) EDCTP-RegNET2015-1104, and 2) Fogarty International Center, National Institutes of Health (grant # 2D43TW009771-06 "HIV and co-infections in Uganda"

"Support for data collection was provided by EDCTP TB NODE, grant number: 1) EDCTP-RegNET2015-1104, and 2) Fogarty International Center, National Institutes of Health (grant # 2D43TW009771-06 "HIV and co-infections in Uganda."

The funders had no role in the study design, data collection, and analysis, decision to publish, or preparation of the manuscript."

4. Please upload a copy of Figures 2-5, to which you refer in your text on page 13 and 15. If the figure is no longer to be included as part of the submission please remove all reference to it within the text.

Additional Editor Comments:

Increased uptake of tuberculosis preventive therapy (TPT) among people living with HIV following the 100-days accelerated campaign: A retrospective review of routinely collected data at six urban public health facilities in Uganda

Musaazi, J et al

Generally, this is an important manuscript touching on a topical issue-uptake of tuberculosis preventive therapy (TPT). I found it generally well written. However, I have a few concerns as listed below.

Comments

Abstract

The Uganda Ministry of Health launched a 100-day campaign to scale-up TPT in PLHIV in July 2019.

Aim: We sought to examine the effect of the campaign on trends of TPT uptake and characteristics associated with TPT uptake and TPT completion among persons in HIV care. We retrospectively reviewed routinely collected data from 2016 to 2019 at six urban public health facilities in Uganda.

Comment 1

If the launch of the campaign was July 2019, and the study reviewed records from 2016 to 2019 at six urban public health facilities in Uganda. Wasn’t it too early to assess the effect of the campaign?

Results

On average a total of 43,215 patients aged 15 years and above were eligible for TPT each 60 calendar year at the six health facilities. More than 70% were females and median age was 34 years 61 (inter-quartile range 28 to 41 years on overall).

Comment 2

This statement is confusing ‘On average a total of 43,215 patients aged 15 years and above were eligible for TPT each 60 calendar year at the six health facilities.’ So the 70% females is of the total eligible patients for all the four years? What was the total eligible patients in the 4 years?

Introduction

Tuberculosis (TB) is the most frequent cause of Acquired Immunodeficiency Syndrome (AIDS)- related deaths worldwide despite the wide availability of antiretroviral therapy (ART)(1). Tuberculosis preventive treatment (TPT) reduces the risk of developing active TB(2) and TB-associated mortality.

Comment 3

In line 78, tuberculosis is abbreviated as ‘TB’ and so subsequently the authors should use the abbreviation ‘TB’ throughout the document eg the beginning of line 79.

Line 102-106 One of the interventions done by the Uganda MoH was the 100-day accelerated isoniazid preventive therapy (IPT) scale-up campaign launched on 3rd 103 July 2019(16). This campaign aimed to enroll 300,000 PLHIV on isoniazid preventive therapy at 1947 ART sites by 30th September 2019. There is scanty published data showing the trends of prevalence of TPT uptake and completion, and the impact of the 2019 Ugandan MoH 100-day TPT scale-up campaign on these trends.

Comment 4

Give the barriers mentioned i.e. ‘inadequate TPT supply in health facilities, frequent drug stock-outs, poor patient adherence, limited TPT knowledge by health workers, lack of confidence in symptom-based TB screening alone, and fear of isoniazid resistance,’ can the authors describe what exactly the entailed this campaign of 100 days to be able to acheive these very impressive results in a very short time July to December, 2022. Considering that the first three months (100 days) could have been invested in the campaign leaving only 3months for implementation and evaluation (including uptake and completion). This means that the time to assess the impact was not long enough and probably these results could be due to some other factors and not necessarily the campaign! The methods section should have this section explaining the details of the campaign/ catch up strategy!

Meanwhile data collection was done in Data were collected from July 2020 to March 2021. (Line 160-161), just wondering why then did evaluate the campaign, a year after its launch, to be able take care of all effects.

Discussion

Comment 5

Line 333-336. Although it was impossible to explore reasons for poor TPT uptake given the retrospective nature of our study, some of the reasons cited for low TPT uptake highlighted in Kalema et.al study included: limited capacity of clinicians to exclude TB using symptoms alone, fear of promoting drug resistance due to isoniazid monotherapy and inconsistent TPT drug supplies(17). Can the authors explain how these barriers were overcame by the 100-days campaign!

Conclusion

Comment 6

Some of the issue raised in the conclusion are not supported by this study’s findings, for example; ). Line 373-374. ‘Also, there is need for constant refresher trainings for health workers to understand importance of 374 TPT, and consistent TPT supply at health facilities.’

Need to recognize the limitation of short time period to evaluate the campaign.
---

## [Author Response · Author response to Decision Letter 0]

2 Aug 2022

RE: RESPONSE TO REVIEWERS’ COMMENTS MANUSCRIPT ID: PONE-D-22-13680 “Increased uptake of tuberculosis preventive therapy (TPT) among people living with HIV following the 100-days accelerated campaign: A retrospective review of routinely collected data at six urban public health facilities in Uganda”. 

The authors of the reviewed manuscript thank the academic editor and reviewers for their consideration and opinions on the manuscript. We particularly thank the reviewers for their careful review of the manuscript and the useful comments. Below, we provide detailed point-by-point responses to the reviewers’ comments in italics and underlined. We have also tracked the changes in the revised manuscript.

ACADEMIC EDITORS’ COMMENT

1: Please ensure that your manuscript meets PLOS ONE's style requirements, including those for file naming. 

Response: The revised manuscript has been reformatted according to PLoS One’s style requirements. 

2: Thank you for stating the following in the Funding Section of your manuscript: 

"Support for data collection was provided by EDCTP, grant number: 1) EDCTP-RegNET2015-1104, and 2) Fogarty International Center, National Institutes of Health (grant # 2D43TW009771-06 "HIV and co-infections in Uganda"

"Support for data collection was provided by EDCTP TB NODE, grant number: 1) EDCTP-RegNET2015-1104, and 2) Fogarty International Center, National Institutes of Health (grant # 2D43TW009771-06 "HIV and co-infections in Uganda."

The funders had no role in the study design, data collection, and analysis, decision to publish, or preparation of the manuscript."

 Response: Thank you for the comment. The funding-related text has been removed from the manuscript. 

The funding statement remains as, "Support for data collection was provided by EDCTP TB NODE, grant number: 1) EDCTP-RegNET2015-1104, and 2) Fogarty International Center, National Institutes of Health (grant # 2D43TW009771-06 "HIV and co-infections in Uganda." 

It was also stated in the online initial submission.

Response: Thank you for the guidance. 

There are no restrictions, and anonymized data set was uploaded at the initial submission. 

4. Please upload a copy of Figures 2-5, to which you refer in your text on page 13 and 15. If the figure is no longer to be included as part of the submission please remove all reference to it within the text.

Response: The figures (2-5) have been uploaded. 

 

Response: The reference list has been reviewed for completeness and correctness. 

ADDITIONAL EDITOR COMMENTS:

Increased uptake of tuberculosis preventive therapy (TPT) among people living with HIV following the 100-days accelerated campaign: A retrospective review of routinely collected data at six urban public health facilities in Uganda 

Musaazi, J et al

 Generally, this is an important manuscript touching on a topical issue- tuberculosis preventive therapy (TPT). I found it generally well written. However, I have a few concerns as listed below. 

Comments 

Abstract 

The Uganda Ministry of Health launched a 100-day campaign to scale-up TPT in PLHIV in July 2019. Aim: We sought to examine the effect of the campaign on trends of TPT uptake and characteristics associated with TPT uptake and TPT completion among persons in HIV care. We retrospectively reviewed routinely collected data from 2016 to 2019 at six urban public health facilities in Uganda. 

Comment 1 

If the launch of the campaign was July 2019, and the study reviewed records from 2016 to 2019 at six urban public health facilities in Uganda. Wasn’t it too early to assess the effect of the campaign? 

Response: We thank the reviewer for the comment. 

The main aim of the 100-day campaign was to initiate TPT for 300,000 adults and children living with HIV and under 5 years contacts of TB patients and ensure 100% completion rates (by 30th September 2019) of individuals that initiated 6-months TPT dose in quarters: October - December 2018, and January-February 2019. 

We, therefore, believe that our assessment of the effect of this campaign at end of December 2019 was very timely and not too early.

However, a limitation of a short-time period evaluation of the campaign has also been added. 

RESULTS 

On average a total of 43,215 patients aged 15 years and above were eligible for TPT each 60 calendar year at the six health facilities. More than 70% were females and median age was 34 years 61 (inter-quartile range 28 to 41 years on overall). 

Comment 2 

This statement is confusing ‘On average a total of 43,215 patients aged 15 years and above were eligible for TPT each 60 calendar year at the six health facilities.’ So the 70% females is of the total eligible patients for all the four years? What was the total eligible patients in the 4 years? 

Response: We agree with the reviewer that the statement was a bit unclear. In the initial statement, we intended to provide a precise summary of the characteristics across the 4 years studied. Which we probably did not phrase very well. The statement has now been rephrased to read as below.

On average a total of 43,215 patients aged 15 years and above were eligible for TPT each calendar year at the six health facilities. Across all the 4 years, more than 70% were females (range: 73.5% -74.6%) and median age ranged from 33 to 34 years.”

Introduction 

Tuberculosis (TB) is the most frequent cause of Acquired Immunodeficiency Syndrome (AIDS)- related deaths worldwide despite the wide availability of antiretroviral therapy (ART)(1). Tuberculosis preventive treatment (TPT) reduces the risk of developing active TB(2) and TBassociated mortality. 

Comment 3 

In line 78, tuberculosis is abbreviated as ‘TB’ and so subsequently the authors should use the abbreviation ‘TB’ throughout the document eg the beginning of line 79. 

Response: We thank the reviewer for the comment. This has been addressed.

We have subsequently used the abbreviation for Tuberculosis “TB” throughout the document after the first occurrence in full name and abbreviation.

Line 102-106 One of the interventions done by the Uganda MoH was the 100-day accelerated isoniazid preventive therapy (IPT) scale-up campaign launched on 3rd 103 July 2019(16). This campaign aimed to enroll 300,000 PLHIV on isoniazid preventive therapy at 1947 ART sites by 30th September 2019. There is scanty published data showing the trends of prevalence of TPT uptake and completion, and the impact of the 2019 Ugandan MoH 100-day TPT scale-up campaign on these trends. 

Comment 4 

Give the barriers mentioned i.e. ‘inadequate TPT supply in health facilities, frequent drug stock-outs, poor patient adherence, limited TPT knowledge by health workers, lack of confidence in symptom based TB screening alone, and fear of isoniazid resistance,’ can the authors describe what exactly the entailed this campaign of 100 days to be able to achieve these very impressive results in a very short time July to December, 2022. Considering that the first three months (100 days) could have been invested in the campaign leaving only 3months for implementation and evaluation (including uptake and completion). This means that the time to assess the impact was not long enough and probably these results could be due to some other factors and not necessarily the campaign! The methods section should have this section explaining the details of the campaign/ catch up strategy! Meanwhile data collection was done in Data were collected from July 2020 to March 2021. (Line 160-161), just wondering why then did evaluate the campaign, a year after its launch, to be able take care of all effects. 

Response: We thank the reviewer for the comment. 

The actual implementation of rapid IPT initiation and enhanced IPT completion started immediately after the launch of the 100-day campaign. We also believe that during the period 3rd July 2019 to 30th September 2019, the 100-day accelerate IPT scale-up was the only intervention conducted in health facilities studied. The impressive impact of the campaign observed in our evaluation could be because the campaign addressed most of the initial barriers to TPT initiation and completion. For example, 1) through enhanced systems for TPT delivery, monitoring, and reporting outcomes, 2) mobilized a DHO-LED multi-stakeholder engagement for accelerated IPT implementation and increased accountability, e.t.c.

We had a statement in the first paragraph of the discussion section explaining the details of the campaign/catch-up strategy. However, we have further listed other activities that were done during the 100-day campaign that could have helped to achieve the success we observed.

Discussion 

Comment 5 

Line 333-336. Although it was impossible to explore reasons for poor TPT uptake given the retrospective nature of our study, some of the reasons cited for low TPT uptake highlighted in Kalema et.al study included: limited capacity of clinicians to exclude TB using symptoms alone, fear of promoting drug resistance due to isoniazid monotherapy and inconsistent TPT drug supplies(17). Can the authors explain how these barriers were overcame by the 100-days campaign! 

Response: Statements have been added on how barriers to TPT uptake and completion were addressed in the 100-day accelerated TPT scale-up campaign. Refer to paragraph #2 in the discussion section.

Conclusion 

Comment 6 

Some of the issue raised in the conclusion are not supported by this study’s findings, for example; ). Line 373-374. ‘Also, there is need for constant refresher trainings for health workers to understand importance of 374 TPT, and consistent TPT supply at health facilities.’ 

Response: The statement has been removed. 

Need to recognize the limitation of short time period to evaluate the campaign.

Response: A limitation of a short-time period evaluation of the campaign has been added.

---

## [Decision Letter · Decision Letter 1]

10 Oct 2022

PONE-D-22-13680R1Increased uptake of tuberculosis preventive therapy (TPT) among people living with HIV following the 100-days accelerated campaign: A retrospective review of routinely collected data at six urban public health facilities in UgandaPLOS ONE

Dear Dr. Musaazi,

Thank you for submitting your manuscript to PLOS ONE. After careful consideration, we feel that it has merit but does not fully meet PLOS ONE’s publication criteria as it currently stands. Therefore, we invite you to submit a revised version of the manuscript that addresses the points raised during the review process.

We look forward to receiving your revised manuscript.

Kind regards,

Lukas Fenner, MD, MSc

Academic Editor

PLOS ONE

Additional Editor Comments:

Please note that I was asked to step in as academic editor (invitation accepted on 13/09/2022) since the initial editor was no longer available. In the interest of a rigorous peer review process, the manuscript was sent out to four independent external reviewers. They made important recommendations and the manuscript cannot be accepted before this revision. Please carefully respond to the reviewers' comments in a detailed point-by-point reply (also included revised text/inserts), and pay particular attention to the following points:Revise methods, including study population and study design, data extraction (including statement on double entry) (see all Reviewers)Missing values (see also Reviewer 2): provide details according to STROBE statement, clearly present missing valuesEvaluation time 3 months after (Reviewer 2): explain and/or provide alternative analysisRevisit analyses and provide a more granular presentation of the data (by quarters, stratified by eligibility criteria for TPT, etc.), 2019 vs. all other years (see all 4 Reviewers)Clear description of the interventions (see Reviewer 1)Expand discussions (potential benefits and harms of such a campaign; see Reviewer 1)Clarify if other interventions took place and could be responsible for the increased uptake (see Reviewer 4)

Reviewers' comments:

Reviewer's Responses to Questions

**Comments to the Author**

1. If the authors have adequately addressed your comments raised in a previous round of review and you feel that this manuscript is now acceptable for publication, you may indicate that here to bypass the “Comments to the Author” section, enter your conflict of interest statement in the “Confidential to Editor” section, and submit your "Accept" recommendation.

Reviewer #1: (No Response)

Reviewer #2: (No Response)

Reviewer #3: (No Response)

Reviewer #4: (No Response)

2. Is the manuscript technically sound, and do the data support the conclusions?

Reviewer #1: Partly

Reviewer #2: Yes

Reviewer #3: Partly

Reviewer #4: Yes

3. Has the statistical analysis been performed appropriately and rigorously? 

Reviewer #1: No

Reviewer #2: Yes

Reviewer #3: I Don't Know

Reviewer #4: Yes

4. Have the authors made all data underlying the findings in their manuscript fully available?

Reviewer #1: No

Reviewer #2: Yes

Reviewer #3: (No Response)

Reviewer #4: (No Response)

5. Is the manuscript presented in an intelligible fashion and written in standard English?

Reviewer #1: Yes

Reviewer #2: Yes

Reviewer #3: (No Response)

Reviewer #4: Yes

6. Review Comments to the Author

**Reviewer #1:** Thank you for asking me to review this manuscript. Although data are interesting, there is currently insufficient methodological and reporting detail in the manuscript to assess what actually happened. I note that in the previous round of reviews, the Editor has already asked for much of this information to be provided. I have provided detailed comments below, and hope the authors are able to fully address them.

1. Lines 79-81 describe the Uganda programme approach to TPT. However, under these guidelines (as described), all PLHIV would take TPT intermittently for the remainder of their lives. Are there time-based criteria for initiation/reinitiating? In practice, what clinical event triggers initiation/reinitiation of TPT (HIV diagnosis/ART initiation etc)? A more careful description in the Methods of the programmatic approach as implemented, beyond what is listed in the guidelines, would be helpful to understand the approach to TPT provision.

2. Methods: add text to provide rationale for the selection of the 6 study HIV clinics, with particular attention to that characteristics of these clinics that mean that data can be generalised from them to the remainder of clinics in the country, and regionally.

3. Lines 106-111: Describe which body in Uganda has responsibility for purchasing, distribution and monitoring of TPT (HIV Programme, TB Programme, other, mixture?)

4. Lines 113-115: Inclusion and exclusion criteria are insufficiently detailed to allow readers to tell who results are applicable to. For example, as currently written, was an HIV-positive person taking ART for 15 years eligible to initiate IPT before the campaign, and during the campaign (to take a somewhat extreme example). If the eligibility criteria for initiation were substantially wider during the campaign, then of course more people are going to be “available” to be initiated onto IPT. However, in this case, the conclusion would be that a campaign is not needed, rather programmes should focus on changing eligibility criteria to boost numbers. Please clearer description of what actually happened before and during the campaign to allow readers to understand what the drivers of success were.

5. Further to point above, is there any evidence to suggest that repeated/continuous courses of IPT are beneficial and not harmful to PLHIV? From figure 1, seems like people more than likely had more than one course of IPT. Was this supported by the national or WHO guidelines. Otherwise, I guess a programme wanting to increase “IPT delivery” could just repeatedly provide IPT to the same set of easier to reach people.

6. Lines 117-124: Similar to points above - need to provide a detailed description of the procedures for assessment and initiation of TPT, not just what happened after initiation. The whole point of this study is to investigate whether a “campaign” increased TPT initiation, but we have essentially no information about what happened before the campaign, or during the campaign. I note that this information was previously asked for in the previous round of reviews, but has again not been provided here.

7. Line 117: which “patients”?

8. Lines 126- needs much more information about the data extraction from paper records, and associated extraction from electronic pharmacy records. What data were extracted? How was quality of extraction assured? How was record linkage between paper and electronic records done and quality assured? How was missing data handled? Also needs key outcome definitions to be provided here (i.e. how was “TPT initiation” defined?)

9. Figure 1. Not clear if individuals are represented more than once in the diagram. I.e. if someone initiated IPT in 2016, would they additionally be eligible to initiate IPT in 2017 (and 2018, and 2019)? Again this comes back to the point about who is actually eligible to start TPT

10. Table 1: I am really unclear about “Total visited and eligible” and “Characteristics of eligible patients for TPT at beginning of each year.” What does “visited” mean? Are these unique people, or person-visits? What happened to people who were diagnosed with HIV/initiated ART (i.e. entered the cohort) during the year?

11. Table 1: I really struggle to believe that, in 2016, no people had previously been diagnosed with TB (especially given the data for the other years). Is this not a data collection issue?

12. Table 1: In line with EQUATOR Guidelines, remove p-values. Given the large sample sizes here, these are probably not very meaningful.

13. How did uptake vary by clinic? What were the drivers and successes of differential uptake by clinic? Was this related to implementation of the campaign?

14. Line 215-217: Again, what is the denominator here. As written, could be either unique participants, or clinic visits (with participants potentially having multiple visits). This really needs to be carefully and correctly described.

15. Although the impact of the campaign seems obvious, it would be more clearly reported if the authors could provide a figure (line-chart) showing the quarterly numbers of TPT initiations over the study period, with the campaign period indicated. As the campaign only during the last study year, this figure would then have sufficient resolution to show the impact of the campaign.

16. Lines 238-239: “Stratified analysis indicated TPT uptake was similar across age groups and sex of participants over the 4 years studied (Fig 2a and 2b)”. This is not what is shown by the regression analysis in Table 2, with adolescents aged 15-19 years having substantially higher IPT uptake compared to older aged people.

17. Table 3: There are substantial missing data here, both for outcomes and denominators. In the methods, the authors state that they used multiple imputation by chained equations, but this doesn’t seem to be the case? Given the risk of misclassification bias, the authors should report complete case analysis, as well as imputed analysis and best/worst case scenario analysis for handling missing data.

18. Lines 204-306: “This was achieved using existing health care resources, system strengthening, multi-stakeholder engagement in the campaign, and enhanced TPT delivery, monitoring and reporting”. This is unfortunately insufficiently clear. Readers planning similar interventions would not be able to replicate the success of the Uganda programme without considerably more detail about what the campaign actually involved.

19. Discussion needs much greater description (likely 1-2 paragraphs) of the potential benefits and harms to individuals and programmes should this “campaign” approach be adopted routinely in Uganda and elsewhere.

20. In the response to the Editor, the authors state that dataset and code for replication are available as Supplemental Material, in line with PLOS One requirements - they are not.

Minor comments

1. Line 33: reword “reactivation, probably to “disease incidence” or similar

2. Figure 1: exclusions might be better show using branching boxes, to add clarity to the flow of the diagram.

3. Line 214 (and throughout next paragraph): “Prevalence” is probably not correct here. Suggest remove to read “Trends in TPT uptake”

**Reviewer #2:** This paper reports on the impact of Uganda's 100-day campaign to scale-up TPT in PLHIV. The results are impressive. The authors have responded to the previous reviewers' comments. I have a few additional comments/questions for the authors to consider:

- Study population: It would be helpful to clarify the inclusion/exclusion criteria a bit more. Would a patient have been included if they visited a clinic multiple times within a year if they were eligible for TPT at one visit but ineligible at a subsequent visit? For example, if a PLHIV presented for care and was eligible for TPT, and later that year returned to clinic and was ineligible (e.g., diagnosed with TB), would they have been included in the analysis?

- There appears to be a typo in the legend for Figure 1. The number reported to have been diagnosed with TB/on TB treatment is the same for 2016-2019 (n=1,168)

- The numbers "visited and eligible" in Table 1 do not match the numbers eligible in Figure 1.

- Do the authors have an explanation for the decline in TPT uptake in 2018? This would be interesting to include in the discussion.

- The authors state "as expected in 2019 uptake increased dramatically by about 25% (relative increase) from 2.5% in 2018 to 64.8% in 2019." I find this confusing -- the increase from 2.5% to 64.8% is many-fold. Could the authors clarify?

- Statement that "TPT uptake was similar across age groups and sex of participants over the 4 years studied" seems contradicted by the data presented in Table 2.

- The discussion states that the evaluation was done 3 months after implementation of the 100-day TPT campaign. Doesn't this mean it was too early to evaluate TPT completion (which takes 6-months) for much of the 2019 study population?

**Reviewer #3: **Thank you for the opportunity to review this manuscript. The authors present findings from an evaluation of a TPT uptake campaign in Uganda. Major revisions are necessary before I would recommend this manuscript for publication.

Major comments:

- Line 98 indicates that this is a ‘cross-sectional review’ – however the statistical analysis is for a longitudinal study design and line 160 notes that there were ‘repeated observation of participants over the 4 years.’ I think cross-sectional isn’t the correct terminology for this paper. Would suggest ‘retrospective cohort study.’

- I would strongly recommend a more granular presentation of this data into quarters or months – given that the 100 day campaign was Q3 of 2019 (July-Sept), it would be informative to see if TPT uptake immediately declined in Q4 or if high levels of TPT prescribing were maintained

- It’s unclear why January 2016 was chosen as the starting point of the study implementation period. Can the authors provide further insight into why the specific time frame was chosen?

- Line 113 states that the study included all ‘asymptomatic’ PLHIV aged 15 years and above. How was symptom status determined? Were the presence or absence of symptoms always recorded (i.e. if a patient had no symptoms, was it clearly stated in the medical record? Or was the absence of documentation of symptoms interpreted as an indication of no symptoms?)? If a patient had multiple visits over a 1 year period and were asymptomatic at the first visit but symptomatic at visit 2, were they included? This could be a major source of misclassification if symptom status was not well documented.

- Lines 189-190 indicate that data were missing for 10% and 4% of patients for TPT uptake and completion, respectively. However, in Figure 1, 26,478 patients initiated on IPT in 2019 but completion was assessed for only 7,713 which would mean that completion data is missing for more than 70% of those who initiated IPT in that year. This raises concerns regarding the validity of the completion analyses and results presented. The authors should provide further detail into why 70% of patients who initiated IPT in 2019 could not be found in the EMR.

- Line 314 of the discussion states that the national guidance prior to 2018 was to initiate only the newly enrolled PLHIV on TPT. This is an important point as that indicates that only new ART initiates were eligible for TPT prior to that change. Figure 1 should be stratified by new vs stable ART clients in order to present TPT uptake based on ART status.

Minor comments:

- The abstract needs section headers (introduction, methods, results, conclusion)

- Lines 77-79 reference two different WHO global TB reports (2018 and 2020). For consistency, please use numbers/figures from the 2020 report. You can consider re-phrasing this section as, “Uganda is among the WHO’s 30 high TB/HIV burden countries which contribute about 60% of the total TB/HIV burden globally. In 2019, approximately XX% of notified TB patients were living with HIV and 6.7% of PLHIV newly enrolled in care were diagnosed with TB. [reference]”

- Lines 79-81: when did the Uganda MoH start recommending TPT for PLHIV? Please add the year to this statement. For example, you could re-phrase as: “Since 20XX, the Uganda Ministry of Health (MoH) has recommended that all PLHIV without symptoms suggestive of active TB disease receive TPT regardless of CD4 count, ART status, history of TB treatment, and pregnancy status.”

- Lines 100-101 describe the 6 health facilities from which patient medical records were abstracted and notes that 5 were health center level III and 1 was a health center IV. Can you add additional detail about what it means for a facility to be level 3 vs level 4? Are there different services provided? Different operating hours? Different staffing levels and/or cadres of staff (i.e. nurses only vs nurses and doctors?)?

- Line 142 states that the data collection tools are presented in the appendix but I couldn’t find them. Please upload them or if they’re not available, remove this sentence.

- The box at the bottom of Figure 1 indicates reasons for patients not being eligible for IPT and lists the exact same number of people (1,168) for each year as diagnosed with active TB or on TB treatment. Please review those numbers for accuracy.

- Line 218 indicates that there was a 25% relative increase in TPT uptake from 2018 to 2019 but it is a 25 times increase, not 25%.

**Reviewer #4:** This study reports the change in the TPT uptake in 6 Uganda's clinics after a nation-wide campaign for TPT upscale in Uganda. The results are very impressive, and I enjoyed really reading the manuscript.

Comments:

1. I appreciate authors using the GEE model that accounts for correlated observations both within clinic and between the same patients across years.

2. Introduction (line 82): authors report TPT uptake of 16-17% before 2019, but their study results (line 217) show only 2.5-4.5%. I wonder whether the patients in the sampled clinics are somehow different from general population, or this is not the same metric that being reported.

3. Introduction (line 86): Authors state the campaign was only one of interventions. What were the others, and could they be also responsible for the increase in the TPT uptake?

4. Methods (line 98): this is not a cross-sectional study. I think it would be best described as a (living) cohort study.

5. Methods (line 100): It should be clarified whether (if) these 6 facilities were all part of the campaign.

6. Methods, outcome: TPT uptake - prevalence vs. incidence. I found the usage of the phrase "prevalence of TPT uptake" a bit misleading, as prevalence typically refers to proportion of all patients with event and "uptake" typically refers to new cases (eg as in incidence). I would recommend authors to make it more clear that the outcome was newly initiated TPT in each year (if that was indeed the case).

7. Methods, model: As there is a natural yearly fluctuation in TPT uptake, authors could consider comparing the TPT uptake in 2019 versus all previous years (2016-2018) combined, rather than comparing to individual years; 2016 (line 220), or 2018 (line 221).

8. Results (line 218): reported change in uptake from 2.5% to 64.8% does not correspond to 25% increase, but rather a 26-fold increase. It is then later stated correctly (eg line 221).

9. Typo (line 54): should be 82.6%

7. PLOS authors have the option to publish the peer review history of their article (what does this mean?). If published, this will include your full peer review and any attached files.

Reviewer #1: No

Reviewer #2: No

Reviewer #3: No

Reviewer #4: No

---

## [Author Response · Author response to Decision Letter 1]

24 Nov 2022

The Editor, 

PLoS One 

Date: 24th November 2022

Dear Sir, 

RE: RESPONSE TO REVIEWERS’ COMMENTS MANUSCRIPT ID: PONE-D-22-13680R1 “Increased uptake of tuberculosis preventive therapy (TPT) among people living with HIV following the 100-days accelerated campaign: A retrospective review of routinely collected data at six urban public health facilities in Uganda”. 

The authors of the reviewed manuscript thank the academic editor and reviewers for their consideration and careful review of the manuscript. Below, we provide detailed point-by-point responses to the reviewers’ comments in italics and underlined. We have also tracked the changes in the revised manuscript.

ACADEMIC EDITORS’ COMMENT

Please note that I was asked to step in as academic editor (invitation accepted on 13/09/2022) since the initial editor was no longer available. In the interest of rigorous peer review process, the manuscript was sent out to four independent external reviewers. They made important recommendations and the manuscript cannot be accepted before this a revision. Please carefully respond to the reviewers' comments in a detailed point-by-point reply (also included revised text/inserts), and pay particular attention to the following points:

1. Revise methods, including study population and study design, data extraction (including statement on double entry) (see all Reviewers)

2. Missing values (see also Reviewer 2): provide details according to STROBE statement, clearly present missing values

3. Evaluation time 3 months after (Reviewer 2): explain and/or provide alternative analysis

4. Revisit analyses and provide a more granular presentation of the data (by quarters, stratified by eligibility criteria for TPT, etc.), 2019 vs. all other years (see all 4 Reviewers)

5. Clear description of the interventions (see Reviewer 1)

6. Expand discussions (potential benefits and harms of such a campaign; see Reviewer 1)

7. Clarify if other interventions took place and could be responsible for the increased uptake (see Reviewer 4)

REVIEWERS' COMMENTS:

Reviewer #1: Thank you for asking me to review this manuscript. Although data are interesting, there is currently insufficient methodological and reporting detail in the manuscript to assess what actually happened. I note that in the previous round of reviews, the Editor has already asked for much of this information to be provided. I have provided detailed comments below, and hope the authors are able to fully address them.

1. Lines 79-81 describe the Uganda programme approach to TPT. However, under these guidelines (as described), all PLHIV would take TPT intermittently for the remainder of their lives. Are there time-based criteria for initiation/reinitiating? In practice, what clinical event triggers initiation/reinitiation of TPT (HIV diagnosis/ART initiation etc)? A more careful description in the Methods of the programmatic approach as implemented, beyond what is listed in the guidelines, would be helpful to understand the approach to TPT provision.

Response: Thank you for the comment. 

According to Ugandan HIV/TB guidelines TPT is only provided to all eligible individuals once in a lifetime and not intermittently. Once the TPT course is completed, the patient does not require another dose. Currently TPT is given to all new PLHIV and anyone in HIV care who never completed or interrupted a course of TPT. Also, patients on ART refill visits are always re-evaluated for TPT history and screened for TB before they are initiated on TPT. 

We have edited the statement in the introduction section that highlights Uganda’s ministry of health TPT programme, and indicated that TPT is offered as a one course of treatment (see page 5, lines 77-79). Also, under setting in the methods section, we have further indicated that during the period studied, Isoniazid monotherapy for six months was the only TPT option available (see page 7, lines 118-120).

We have added more statements in the methods section (see page 7, lines 126 – 130 and page 9, lines 171-178) to highlight the programmatic approach as implemented, beyond what is listed in the guidelines. For example, in programmatic setting, depending on the stability of the patient in terms of health status and ART adherence, ART medications can be provided for either a month, or 3 months or 6 months. Therefore, TPT refill visits are aligned with ART refill visits in order to decrease the number of patients visits to the clinic which can be costly to the patient in terms of transport and time.

2. Methods: add text to provide rationale for the selection of the 6 study HIV clinics, with particular attention to that characteristics of these clinics that mean that data can be generalised from them to the remainder of clinics in the country, and regionally.

Response: We have clarified the rationale for selection of the 6 study HIV clinics and briefly described the characteristics of these clinics (see pages 6, lines 103-116). 

About generalisability, since Uganda is a limited-resource country and has a high burden of HIV and TB coinfections which is more pronounced in urban areas like Kampala than in rural areas, our findings can be generalisable to other similar resource-limited settings with high burden of TB and HIV coinfections. We included this under discussion section (see page 23, lines 438-439).

3. Lines 106-111: Describe which body in Uganda has responsibility for purchasing, distribution and monitoring of TPT (HIV Programme, TB Programme, other, mixture?)

Response: In Uganda, TPT is procured under the Quantification planning and procurement Unit (QPPU) of the ministry of health. Distribution of TPT is done by the government national warehouse (National medical stores, i.e. NMS) while monitoring and support supervision is done by Ministry of Health under the National TB and leprosy program (NTLP).

The above statement has been added under setting in methods section (see page 7, lines 130-133)

4. Lines 113-115: Inclusion and exclusion criteria are insufficiently detailed to allow readers to tell who results are applicable to. For example, as currently written, was an HIV-positive person taking ART for 15 years eligible to initiate IPT before the campaign, and during the campaign (to take a somewhat extreme example). If the eligibility criteria for initiation were substantially wider during the campaign, then of course more people are going to be “available” to be initiated onto IPT. However, in this case, the conclusion would be that a campaign is not needed, rather programmes should focus on changing eligibility criteria to boost numbers. Please clearer description of what actually happened before and during the campaign to allow readers to understand what the drivers of success were.

Response: The same national TPT guidelines were followed before and after the accelerated IPT campaign: All people living with HIV were eligible after excluding active TB disease, regardless of their duration on ART. However, during the 100-days campaign in 2019, the Uganda Ministry of Health aimed at achieving a rapid IPT initiation and completion during that period. Besides, the Ministry wanted to jump-start the TPT program and also address some challenges in the country TPT program e.g., limited isoniazid availability at health facilities due to supply chain challenges, lack of awareness of the potential benefits of isoniazid; health facilities, district & implementing partners not being held accountable for IPT performance.

In order to make it clear to the reader what happened before and during the campaign, we have added statements under setting in methods section describing the activities conducted during the 100-day accelerated IPT campaign (see page 8, lines 134-146).

We further provide the reference for more details about this campaign, i.e. reference #18: Uganda Ministry of Health. 100-Day Accelerated Isoniazid Preventive Therapy Scale Up Plan, 2019.

5. Further to point above, is there any evidence to suggest that repeated/continuous courses of IPT are beneficial and not harmful to PLHIV? From figure 1, seems like people more than likely had more than one course of IPT. Was this supported by the national or WHO guidelines. Otherwise, I guess a programme wanting to increase “IPT delivery” could just repeatedly provide IPT to the same set of easier to reach people.

Response: Thank you for pointing out the need to clarify this point. There is no recommendation in place for repeated TPT. According to the Uganda Ministry of Health TPT guideline that applied during the period when the data was collected, people living with HIV who had never previously received IPT and did not have evidence of active TB received IPT. 

Figure 1 has a footnote showing the number of patients who were excluded from the analysis in a specific year because of IPT history. We excluded both newly enrolled in the clinic with prior history of IPT uptake and the continuing patients initiated on IPT in the previous year.

To make this clearer, we have added a statement under the eligibility section indicating “Individuals who were initiated on TPT in the previous calendar year were excluded in the analysis of current year.” (see page 8, lines 153-157).

6. Lines 117-124: Similar to points above - need to provide a detailed description of the procedures for assessment and initiation of TPT, not just what happened after initiation. The whole point of this study is to investigate whether a “campaign” increased TPT initiation, but we have essentially no information about what happened before the campaign, or during the campaign. I note that this information was previously asked for in the previous round of reviews, but has again not been provided here.

Response: Thank you for the comment.

The Uganda Ministry of Health has recommended TPT since 2006 under the national policy guidelines for TB/HIV collaborative activities [14,] but due to challenges with isoniazid stocks (because TB treatment is in fixed dose combinations), the number of people receiving TPT is limited to when a full six month course can be supplied. During the 100 days campaign, a significant stock of TPT was received and pushed to the facilities by National Medical Stores to initiate all eligible PLHIV. Prior to initiation of TPT, PLHIV were first screened for signs and symptoms of TB using the TB case finding form and any presumptive case would be first investigated to rule out TB before TPT initiation; those without TB symptoms would then be screened further to rule out any contraindication to isoniazid prior initiation to TPT (see S2 table X2 in supplementary 2, the checklists for screening for contraindications).

The above statement has been added under treatment procedure section in methods (see page 9, lines 165-181 and page 10 lines 182-184). 

7. Line 117: which “patients”?

Response: Thank you for the comment.

We have corrected the statement to indicate that “PLHIV” (see page 9, lines 165-167).

8. Lines 126- needs much more information about the data extraction from paper records, and associated extraction from electronic pharmacy records. What data were extracted? How was quality of extraction assured? How was record linkage between paper and electronic records done and quality assured? How was missing data handled? Also needs key outcome definitions to be provided here (i.e. how was “TPT initiation” defined?)

Response: Thank you for the comment.

Responding to each individual question in this comment.

 What data were extracted? 

A list of data variables has been added under methods section in the data extraction and collection (see page10, lines 191-197). Also, data collection tool is attached as a separate supplementary material. 

How was quality of extraction assured?

Under methods in the data extraction and collection section, we have added a statement that explains how data quality was assured as follows: “A structured questionnaire for extraction of TPT information from the paper-based TPT register was designed and programmed into Open Data Kit (ODK) , a mobile data collection application. Data quality was controlled by applying data validation checks in the ODK database system, training of research assistants who extracted data from TPT registers and continuous review of data captured into ODK.” (see page 10, lines 198-213).

How was missing data handled?

We added the following statement under data extraction section on how we handled missing or unlinked patients’ identifiers: “TPT data was linked to HIV care data using the patient HIV clinic identifier number (i.e. IDCNO). If the identifier was missing in the TPT paper-based register, then patient characteristics including; names, age, sex, home address, ART start date were used if available.” (see page 10, lines 202-204).

Also, under “Evaluation of TPT uptake and outcomes” section, we indicated that “…we additionally extracted prescribed drugs recorded in EMR to identify more individuals for whom TPT was prescribed during the study period in order to reduce on underreporting IPT uptake due to missing information” (see page 10, lines 186-189).

For handling missing data at analysis, we used a multiple imputation method and worst-case and best-case scenarios as sensitivity analysis approaches; the statement is under the statistical analysis section (see page 12, lines 243-245).

Key outcome definitions to be provided here (i.e. how was “TPT initiation” defined?)

Under statistical analysis section, we provided key outcome definitions for TPT uptake (i.e. TPT initiation) and TPT completion (see page 11, lines 223-226). 

9. Figure 1. Not clear if individuals are represented more than once in the diagram. I.e. if someone initiated IPT in 2016, would they additionally be eligible to initiate IPT in 2017 (and 2018, and 2019)? Again this comes back to the point about who is actually eligible to start TPT

Response: Thank you for the comment.

We have two scenarios highlighted below:

Scenario 1: If an individual had a clinic encounter anywhere between 2016 and in 2017, and was eligible for IPT but was not initiated on IPT in 2016, this individual would be included in the denominator for calculation of the proportion of IPT initiation in both 2016 and 2017

Scenario 2: If an individual had a clinic encounter between 2016 and in 2017, and was eligible for IPT and was initiated on IPT in 2016, this individual would not be counted in 2017.

Figure 1 has a footnote showing the number of patients who were excluded in the analysis for a specific year because they were on IPT prior. This includes both newly enrolled in the clinic and had prior history of IPT uptake and the continuing patients in the clinic who were initiated on IPT in the previous year.

To make this clearer, we have added a statement under the inclusion/exclusion section indicating “Individuals who were initiated on TPT in the previous calendar year were excluded in the analysis of subsequent years.” (see page 8, lines 153-157).

10. Table 1: I am really unclear about “Total visited and eligible” and “Characteristics of eligible patients for TPT at beginning of each year.” What does “visited” mean? Are these unique people, or person-visits? What happened to people who were diagnosed with HIV/initiated ART (i.e. entered the cohort) during the year?

Response: We have corrected the confusing terms on Table 1. 

The statements now read as “Total patients who had at least one clinic encounter during the year and were eligible for TPT”. This also accounts for patients enrolled into HIV care during the year. The counts presented are for unique patients. 

We have also edited the footnote on Table 1 to indicate that “Characteristics of patients eligible for TPT at their first visit at the clinic in a specific year”.

11. Table 1: I really struggle to believe that, in 2016, no people had previously been diagnosed with TB (especially given the data for the other years). Is this not a data collection issue?

Response: Thank you for the comment.

We agree that this is a data collection issue. The number of people previously diagnosed with TB for 2016 cannot be ascertained because data extraction was restricted to the period 2016-2019. We did not extract data prior 2016, thus TB history prior 2016 could not be ascertained. 

We have dropped the TB history variable from the analysis because it could be accurately ascertained. We have updated tables 1, 2, 3 and the text accordingly.

12. Table 1: In line with EQUATOR Guidelines, remove p-values. Given the large sample sizes here, these are probably not very meaningful.

Response: The p-values in Table 1 have been removed.

13. How did uptake vary by clinic? What were the drivers and successes of differential uptake by clinic? Was this related to implementation of the campaign?

Response: Thank you for the comment.

TPT uptake did not significantly vary across clinics before and after the year of 100-days campaign. Implementation of the campaign was similar across the six clinics.

We have added a statement on TPT uptake across clinics in the results section (page 16, lines 289-290) and table provided in supplementary 2 (i.e.S2 Table 3a).

14. Line 215-217: Again, what is the denominator here. As written, could be either unique participants, or clinic visits (with participants potentially having multiple visits). This really needs to be carefully and correctly described.

Response: These were unique participant in each year. A participant could only be carried over and counted in the denominator for the following year if he or she never initiated on TPT in previous year. The statement has been edited to avoid the confusion articulated by the reviewer (see page 16, lines 281-290).

15. Although the impact of the campaign seems obvious, it would be more clearly reported if the authors could provide a figure (line-chart) showing the quarterly numbers of TPT initiations over the study period, with the campaign period indicated. As the campaign only during the last study year, this figure would then have sufficient resolution to show the impact of the campaign.

Response: We agree. A line graph for TPT uptake indicating the campaign period has been plotted and included in supplementary 2, and labelled “S2 Fig 1”. Also, a statement referencing this has been added in the results section (see page 16, line 290).

16. Lines 238-239: “Stratified analysis indicated TPT uptake was similar across age groups and sex of participants over the 4 years studied (Fig 2a and 2b)”. This is not what is shown by the regression analysis in Table 2, with adolescents aged 15-19 years having substantially higher IPT uptake compared to older aged people.

Response: Thank you for the comment.

Stratified analysis was performed by fitting a regression model with interaction between calendar years and selected participants’ characteristics to compare trends of TPT uptake across covariates’ groups overtime. E.g. stratified analysis was to answer a question like, “Was the trend of TPT uptake different across sex over the 4 years?”. Whereas, results presented in Table 2 are for the main effects (i.e. overall effect) of the characteristics on TPT uptake. i.e. estimates from Table 2 can answer a question like “On overall, is there a difference in TPT uptake across sex of the participant?”. So, stratified analysis addresses a question on trend whereas main effects (i.e. Table 2) address overall effect of the covariate.

We have edited the statement on stratified analysis results to indicate that results apply trends over the 4 years studied (see page 17, lines 304).

17. Table 3: There are substantial missing data here, both for outcomes and denominators. In the methods, the authors state that they used multiple imputation by chained equations, but this doesn’t seem to be the case? Given the risk of misclassification bias, the authors should report complete case analysis, as well as imputed analysis and best/worst case scenario analysis for handling missing data.

Response: Thank you for the comment.

Sensitivity analyses for handling missing data i.e. multiple imputation, worst-case and best-case scenarios have been performed and a statement has been added in results section (see page 18, lines 331-335). Also, detailed results on all covariates in models are presented in supplementary 2, table 4.

18. Lines 204-306: “This was achieved using existing health care resources, system strengthening, multi-stakeholder engagement in the campaign, and enhanced TPT delivery, monitoring and reporting”. This is unfortunately insufficiently clear. Readers planning similar interventions would not be able to replicate the success of the Uganda programme without considerably more detail about what the campaign actually involved.

Response: Thank you for the comment.

An additional paragraph explaining what was involved in the campaign has been added in methods section (see page 8, lines 134-146, and page 9, lines 171-184). For more detailed information about the implementation of the 100-days campaign, we provided reference #18 (i.e. document on “100-Day Accelerated Isoniazid Preventive Therapy Scale Up Plan, 2019”). Additionally, in supplementary 2, we provided the outline of key activities and their timelines during the campaign. 

19. Discussion needs much greater description (likely 1-2 paragraphs) of the potential benefits and harms to individuals and programmes should this “campaign” approach be adopted routinely in Uganda and elsewhere.

Response: Thank you for the comment.

An additional paragraph on potential benefits and harms to individuals, has been added in the under the discussion section (see page 20, lines 383-390).

20. In the response to the Editor, the authors state that dataset and code for replication are available as Supplemental Material, in line with PLOS One requirements - they are not.

Response: Thank you for the comment.

The dataset has now been provided as supplementary material, and labelled as “supplementary 1 _tpt_data_PLosOne”.

Minor comments

1. Line 33: reword “reactivation, probably to “disease incidence” or similar

Response: Thank you for the comment.

The statement has been reworded to “Tuberculosis preventive therapy (TPT) effectively decreases rates of developing active tuberculosis disease in people living with HIV (PLHIV) who are at increased risk.” (see page 3, lines 32-34).

2. Figure 1: exclusions might be better show using branching boxes, to add clarity to the flow of the diagram.

Response: Thank you for the comment.

Figure 1 has been edited with exclusions indicated as branching boxes.

3. Line 214 (and throughout next paragraph): “Prevalence” is probably not correct here. Suggest remove to read “Trends in TPT uptake”

 Response: Thank you for the comment.

The paragraph has been edited and term “prevalence” has been removed.

Reviewer #2: This paper reports on the impact of Uganda's 100-day campaign to scale-up TPT in PLHIV. The results are impressive. The authors have responded to the previous reviewers' comments. I have a few additional comments/questions for the authors to consider:

1- Study population: It would be helpful to clarify the inclusion/exclusion criteria a bit more. Would a patient have been included if they visited a clinic multiple times within a year if they were eligible for TPT at one visit but ineligible at a subsequent visit? For example, if a PLHIV presented for care and was eligible for TPT, and later that year returned to clinic and was ineligible (e.g., diagnosed with TB), would they have been included in the analysis?

Response: Thank you for the comment.

If a patient was eligible at start of the current year but during the course of the year became ineligible e.g. diagnosed with TB, this patient could be considered in the denominator for the current year but excluded in denominators in the subsequent years.

Additional statement had been added in the inclusion and exclusion criteria as guided (see page 8, lines 153-157).

2- There appears to be a typo in the legend for Figure 1. The number reported to have been diagnosed with TB/on TB treatment is the same for 2016-2019 (n=1,168)

Response: Thank you for the comment.

The numbers “diagnosed with TB/on TB treatment” have been corrected.

3- The numbers "visited and eligible" in Table 1 do not match the numbers eligible in Figure 1.

Response: Thank you for the comment.

Table 1 Included only participants who appeared in the EMR since it was the source document with detailed participants’ characteristics. 

Number of eligible participants presented in table 1 compared to those in figure 1, are less by number of participants who were in paper-based TB registers but their IDs could not be linked with HIV data in the EMR, i.e. 22, 30, 63, 380 in 2016, 2017, 2018 and 2019 respectively.

A footnote has been added below table 1 with an explanation of this difference. 

4- Do the authors have an explanation for the decline in TPT uptake in 2018? This would be interesting to include in the discussion.

Response: Thank you for the comment.

The decline in TPT uptake in 2018 was due to general stockouts at the National Medical Stores and health facilities were affected.

5- The authors state "as expected in 2019 uptake increased dramatically by about 25% (relative increase) from 2.5% in 2018 to 64.8% in 2019." I find this confusing -- the increase from 2.5% to 64.8% is many-fold. Could the authors clarify?

Response: The increase was ~26 fold comparing TPT uptake in 2018 compared to 2019 (64.8/2.5 = 25.92) and has been reported as such. Thanks for pointing out this error.

6- Statement that "TPT uptake was similar across age groups and sex of participants over the 4 years studied" seems contradicted by the data presented in Table 2.

Response: Thank you for the comment.

Stratified analysis was performed by fitting a regression model with interaction between calendar years and selected participants’ characteristics to compare trends of TPT uptake across covariates’ groups overtime. E.g. stratified analysis was to answer a question like, “Was the trend of TPT uptake different across sex over the 4 years?”. Whereas, results presented in Table 2 are for the main effects (i.e. overall effect) of the characteristics on TPT uptake. i.e. estimates from Table 2 can answer a question like “On overall, is there a difference in TPT uptake across sex of the participant?”. So, stratified analysis addresses a question on trend whereas main effects (i.e. Table 2) address overall effect of the covariate.

We have edited the statement on stratified analysis results to indicate that results apply trends over the 4 years studied (see page 17, lines 304).

7- The discussion states that the evaluation was done 3 months after implementation of the 100-day TPT campaign. Doesn't this mean it was too early to evaluate TPT completion (which takes 6-months) for much of the 2019 study population?

 Response: Thank you for the comment.

The main aim of the 100-day IPT accelerated campaign was to initiate 300,000 PLHIV on IPT from July 2019 to September 2019 and achieve IPT completion rates of 100% of individuals previously initiated on TPT in quarters: October – December 2018, and January – February 2019. So, TPT completion was evaluated only among those who previously initiated on TPT for ≥6 months prior to the campaign.

We have edited statement in the discussion section first paragraph (see page 20, lines 375-379) to clearly show the main aim of the campaign in terms of TPT completion was to achieve 100% TPT completion among those initiated ≥6 months prior the campaign. Additionally, we have edited the definition for TPT completion under statistical analysis section (see page 11, lines 224-226).

Reviewer #3: Thank you for the opportunity to review this manuscript. The authors present findings from an evaluation of a TPT uptake campaign in Uganda. Major revisions are necessary before I would recommend this manuscript for publication.

Major comments:

1- Line 98 indicates that this is a ‘cross-sectional review’ – however the statistical analysis is for a longitudinal study design and line 160 notes that there were ‘repeated observation of participants over the 4 years.’ I think cross-sectional isn’t the correct terminology for this paper. Would suggest ‘retrospective cohort study.’

Response: Thank you for the comment. 

We have re-written the statement and indicated that it was cohort study design that involved retrospective review of patients’ medical records (see page 6, line 101).

2- I would strongly recommend a more granular presentation of this data into quarters or months – given that the 100 day campaign was Q3 of 2019 (July-Sept), it would be informative to see if TPT uptake immediately declined in Q4 or if high levels of TPT prescribing were maintained

Response: Thank you for the comment.

A line graph for TPT uptake in quarters has been plotted and included in supplementary 2, and labeled “S2 Fig 1”. Also, a statement referencing this has been added in the results section (see page 16, line 290).

3- It’s unclear why January 2016 was chosen as the starting point of the study implementation period. Can the authors provide further insight into why the specific time frame was chosen?

Response: Thank you for the comment.

We arbitrarily chose January 2016 as the starting point for our evaluation, because a health workers’ guide for provision of isoniazid preventive therapy to PLHIV was rolled out in June 2014 and we wanted to allow enough time for the information to be disseminated and implemented. Therefore, we reasoned that one and half years after rolling out the IPT guide, health workers would have mastered the guide and the IPT program implementation would running effectively.

An additional statement to reflect this has been added under the inclusion in the methods section (see page 9, lines 158-163).

4- Line 113 states that the study included all ‘asymptomatic’ PLHIV aged 15 years and above. How was symptom status determined? Were the presence or absence of symptoms always recorded (i.e. if a patient had no symptoms, was it clearly stated in the medical record? Or was the absence of documentation of symptoms interpreted as an indication of no symptoms?)? If a patient had multiple visits over a 1 year period and were asymptomatic at the first visit but symptomatic at visit 2, were they included? This could be a major source of misclassification if symptom status was not well documented.

Response: Thank you for the comment.

If a patient was eligible at the start of the current year but during the course of the year (ie. On a subsequent visit within the year) became ineligible e.g. became symptomatic or diagnosed with TB, this patient could be considered in the denominator for the current year but excluded in denominators in the subsequent years. An additional statement had been added in the inclusion and exclusion criteria as guided (see page 8, lines 154-157). 

Given the retrospective nature of the study, we could only use the documented information to classify the presence or absence of symptoms. We agree that there is high likelihood of misclassification especially if symptom status was not documented. This has been acknowledged under limitation.

5- Lines 189-190 indicate that data were missing for 10% and 4% of patients for TPT uptake and completion, respectively. However, in Figure 1, 26,478 patients initiated on IPT in 2019 but completion was assessed for only 7,713 which would mean that completion data is missing for more than 70% of those who initiated IPT in that year. This raises concerns regarding the validity of the completion analyses and results presented. The authors should provide further detail into why 70% of patients who initiated IPT in 2019 could not be found in the EMR.

Response: Thank you for the comment.

The high proportion of unlinked TPT register and HIV care data was because some patients’ HIV clinic identifiers that were used link HIV care and TPT data were either missing or wrongly captured in the paper-based TPT registers. Also, since TPT is only offered for a short time i.e. 6 months for isoniazid monotherapy, and that most patients received a full six-months course at once, it could be hard to reconcile the wrongly entered identifiers in the paper-based IPT registers with those in the ART clinics. However, when we compared the characteristics of patients whose TPT data and HIV care data was linked versus those whose data was not linked, they were similar. This implies that there was little potential effect of selection bias. 

In the manuscript, we explained this in discussions section under limitations (see page 22, lines 429-435).

6- Line 314 of the discussion states that the national guidance prior to 2018 was to initiate only the newly enrolled PLHIV on TPT. This is an important point as that indicates that only new ART initiates were eligible for TPT prior to that change. Figure 1 should be stratified by new vs stable ART clients in order to present TPT uptake based on ART status.

Response: Thank you for the comment.

There is information on TPT uptake stratified by ART status presented in supplementary material 2, table S2 Table 2 Proportion of TPT uptake across participants characteristics by year.

Minor comments:

7- The abstract needs section headers (introduction, methods, results, conclusion)

Response: Thank you for the comment.

According to the PLOS ONE guidelines, there are no section headers in the abstract.

8- Lines 77-79 reference two different WHO global TB reports (2018 and 2020). For consistency, please use numbers/figures from the 2020 report. You can consider re-phrasing this section as, “Uganda is among the WHO’s 30 high TB/HIV burden countries which contribute about 60% of the total TB/HIV burden globally. In 2019, approximately XX% of notified TB patients were living with HIV and 6.7% of PLHIV newly enrolled in care were diagnosed with TB. [reference]”

Response: Thank you for the comment.

The statement has been rephrased and the statistics and reference has been added (see page 5, lines 74-76).

9- Lines 79-81: when did the Uganda MoH start recommending TPT for PLHIV? Please add the year to this statement. For example, you could re-phrase as: “Since 20XX, the Uganda Ministry of Health (MoH) has recommended that all PLHIV without symptoms suggestive of active TB disease receive TPT regardless of CD4 count, ART status, history of TB treatment, and pregnancy status.”

Response: Thank you for the comment. 

The statement has been revised accordingly, and the year when the Uganda MoH started recommending TPT for PLHIV has been indicated (see page 5, lines 77-79). 

10- Lines 100-101 describe the 6 health facilities from which patient medical records were abstracted and notes that 5 were health center level III and 1 was a health center IV. Can you add additional detail about what it means for a facility to be level 3 vs level 4? Are there different services provided? Different operating hours? Different staffing levels and/or cadres of staff (i.e. nurses only vs nurses and doctors?)?

Response: Thank you for the comment. 

We have provided some details on services provided at health facility levels 3 and 4 (see page 7, lines 109-112). 

11- Line 142 states that the data collection tools are presented in the appendix but I couldn’t find them. Please upload them or if they’re not available, remove this sentence.

Response: Thank you for the comment. 

The data collection tools are now provided as supplementary 3.

12- The box at the bottom of Figure 1 indicates reasons for patients not being eligible for IPT and lists the exact same number of people (1,168) for each year as diagnosed with active TB or on TB treatment. Please review those numbers for accuracy.

Response: Thank you for the comment. These were typos, now corrected.

13- Line 218 indicates that there was a 25% relative increase in TPT uptake from 2018 to 2019 but it is a 25 times increase, not 25%.

Response: The increase was ~26 fold comparing TPT uptake in 2018 compared to 2019 (64.8/2.5 = 25.92) and has been reported as such. Thanks for pointing out this error.

 

Reviewer #4: This study reports the change in the TPT uptake in 6 Uganda's clinics after a nation-wide campaign for TPT upscale in Uganda. The results are very impressive, and I enjoyed really reading the manuscript.

Comments:

1. I appreciate authors using the GEE model that accounts for correlated observations both within clinic and between the same patients across years.

Response: Thank you for the comment.

2. Introduction (line 82): authors report TPT uptake of 16-17% before 2019, but their study results (line 217) show only 2.5-4.5%. I wonder whether the patients in the sampled clinics are somehow different from general population, or this is not the same metric that being reported.

Response: Thank you for the comment.

There are number of differences in population definitions between the quoted study (the statistics we present in the introduction) and our study. 

1) Different clinics were studied. 

2) The quoted study included only PLHIV newly registered for ART in 2017 whereas in our study we included both newly enrolled and ART experienced PLHIV. In our study we observed a slightly higher TPT uptake before the 100-day campaign for PLHIV newly enrolled in HIV care/newly started on ART compared to the ART experienced, which seems to agree with the TPT uptake documented in the quoted study. As explained in the discussion, in programmatic setting, clinics tended to prioritise TPT initiation to PLHIV newly enrolled in HIV care than the ART experienced when there was limited TPT stock.

 3) Also, the quoted study only sampled on 400 PLHIV from the selected clinics whereas in our study we analysed had everyone that was eligible for TPT. 

3. Introduction (line 86): Authors state the campaign was only one of interventions. What were the others, and could they be also responsible for the increase in the TPT uptake?

Response: Thank you for the comment.

Yes, the 100-day accelerated Isoniazid preventive therapy scale-up campaign was the only campaign conducted during the period. We therefore believe that the increase in the TPT uptake we observed was due to this campaign.

4. Methods (line 98): this is not a cross-sectional study. I think it would be best described as a (living) cohort study.

Response: Thank you for the comment. We have re-written the statement and indicated that it was cohort study design that involved retrospective review of patients’ medical records (see page 7, line 101).

5. Methods (line 100): It should be clarified whether (if) these 6 facilities were all part of the campaign.

Response: Thank you for the comment.

A sentence has been added under setting in methods section (see page 8, line 143).

6. Methods, outcome: TPT uptake - prevalence vs. incidence. I found the usage of the phrase "prevalence of TPT uptake" a bit misleading, as prevalence typically refers to proportion of all patients with event and "uptake" typically refers to new cases (eg as in incidence). I would recommend authors to make it more clear that the outcome was newly initiated TPT in each year (if that was indeed the case).

Response: Thank you for the comment.

We have edited the definition of TPT uptake to read as newly initiated on TPT among PLHIV in HIV care who were eligible for TPT in a specific year (see page 11, lines 223-224). 

7. Methods, model: As there is a natural yearly fluctuation in TPT uptake, authors could consider comparing the TPT uptake in 2019 versus all previous years (2016-2018) combined, rather than comparing to individual years; 2016 (line 220), or 2018 (line 221).

Response: Thank you for the comment.

We agree that the analysis of combined previous years (i.e. 2016-2018) compared to 2019, would be interesting, however, the aim of the study was to compare trends overtime and the impact of the 100-day campaign on these trends. Also, combining all the previous years (2016-2018) may not account the variations in implementation of the TPT program across the 3 years.

8. Results (line 218): reported change in uptake from 2.5% to 64.8% does not correspond to 25% increase, but rather a 26-fold increase. It is then later stated correctly (eg line 221).

Response: The increase was ~26 fold comparing TPT uptake in 2018 compared to 2019 (64.8/2.5 = 25.92) and has been reported as such. Thanks for pointing out this error.

9. Typo (line 54): should be 82.6%

Response: Thank you for the comment. The typo has been corrected.

Thank you

Joseph Musaazi

Corresponding Author

---

## [Decision Letter · Decision Letter 2]

19 Dec 2022

PONE-D-22-13680R2Increased uptake of tuberculosis preventive therapy (TPT) among people living with HIV following the 100-days accelerated campaign: A retrospective review of routinely collected data at six urban public health facilities in UgandaPLOS ONE

Dear Dr. Musaazi,

Thank you for submitting your manuscript to PLOS ONE. After careful consideration, we feel that it has merit but does not fully meet PLOS ONE’s publication criteria as it currently stands. Therefore, we invite you to submit a revised version of the manuscript that addresses the points raised during the review process.

We look forward to receiving your revised manuscript.

Kind regards,

Lukas Fenner, MD, MSc

Academic Editor

PLOS ONE

Journal Requirements:

Additional Editor Comments:

Many thanks for the first revisions. As seen below, Reviewer 3 has still some critical concerns around the analysis.

Reviewers' comments:

Reviewer's Responses to Questions

**Comments to the Author**

1. If the authors have adequately addressed your comments raised in a previous round of review and you feel that this manuscript is now acceptable for publication, you may indicate that here to bypass the “Comments to the Author” section, enter your conflict of interest statement in the “Confidential to Editor” section, and submit your "Accept" recommendation.

Reviewer #1: All comments have been addressed

Reviewer #2: All comments have been addressed

Reviewer #3: (No Response)

Reviewer #4: All comments have been addressed

Reviewer #5: (No Response)

2. Is the manuscript technically sound, and do the data support the conclusions?

Reviewer #1: Yes

Reviewer #2: Yes

Reviewer #3: Partly

Reviewer #4: Yes

Reviewer #5: Yes

3. Has the statistical analysis been performed appropriately and rigorously? 

Reviewer #1: Yes

Reviewer #2: Yes

Reviewer #3: I Don't Know

Reviewer #4: Yes

Reviewer #5: Yes

4. Have the authors made all data underlying the findings in their manuscript fully available?

Reviewer #1: Yes

Reviewer #2: Yes

Reviewer #3: Yes

Reviewer #4: (No Response)

Reviewer #5: Yes

5. Is the manuscript presented in an intelligible fashion and written in standard English?

Reviewer #1: Yes

Reviewer #2: Yes

Reviewer #3: Yes

Reviewer #4: Yes

Reviewer #5: Yes

6. Review Comments to the Author

Reviewer #1: Thanks to the authors for a comprehensive response, which has strengthened the manuscript. I have no other questions.

Reviewer #2: The authors have responded to all of the queries. I have two final thoughts:

- Line 158: The authors can remove "arbitrarily," which has a negative connotation. There is rationale for the starting point of the evaluation.

- Analysis/Discussion: The TPT completion rates are extremely high. Though promising, I worry this could be partially due to bias from only using the EMR records (which comprised only approximately 30% of the TPT initiation from 2019). The authors address this is the discussion, but conclude that there was unlikely to be bias. I am less certain, and think the results warrant a bit more caution in their interpretation.

Reviewer #3: Thank you for the opportunity to review this revised manuscript. The authors present findings from an evaluation of a TPT uptake campaign in Uganda. Further revisions are necessary before I would recommend this manuscript for publication.

Major comments:

- Given that the short duration of the campaign and the focus on completion for patients initiated in Q4 2018 and Q1 2019, I would suggest that the primary analysis be conducted quarterly rather than annually (as opposed to the quarterly data being presented only in the appendix). Or is there a statistical reason to focus on annual data for the primary analysis?

- On lines 392-394 of the discussion, the authors indicate that national guidance before 2018 was to enroll only the newly enrolled PLHIV in care and TPT eligibility was expanded in 2019. In figure 1, it’s indicated that 38,704 patients were eligible for TPT in 2016, for example. Is this correct? Are these only newly enrolled? If this is all PLHIV, I would suggest that the primary analysis be focused only on newly enrolled PLHIV since they were the only patients eligible until the last year covered by this analysis. Data on established ART clients could be presented in a supplementary appendix.

- Figure 1 and table 1 have slightly different numbers – as per the footnote, table 1 excludes individuals who were in paper based registers but not the EMR. Can figure 1 be revised so that the data across figure 1 and table 1 are consistent?

Minor comments:

- On lines 89-91 (and again on lines 136-137), the authors describe the 100 day campaign and indicate that completion was focused on patients who initiated in the quarters October-December 2018 and January-February 2019. Should the second quarter be January-March 2019 (to reflect a 3 month period?)?

- On line 110, the comma after ‘inpatient health’ should come after the word ‘services’

- Line 256 states that 95% of patients were eligible for TPT during each year. However, in 2018, it was 91% (40,390/44,439) and in 2019, it was 89% (40,867/45,868). Line 256 should be revised to reflect the data presented in figure 1.

- On lines 258-259, the authors state that ‘data were missing on 10% and 4% of patients for TPT uptake and completion analyses, respectively.’ However, 26,478 people initiated TPT in 2019 and only 7,713 were evaluated for completion. So isn’t completion data missing for 71% of individuals who initiated TPT in that year?

- In the response to reviewer #2 (point 4), the authors indicated that the decline observed in 2018 was due to stockouts at the national medical stores but I didn’t see this reported in the revised manuscript. This could be added to the discussion to provide more context for the results.

- While the discussion states that national guidance was to provide TPT to only those newly enrolled in care through 2018, the introduction (lines 77-79) and the methods (lines 120-122) state that the MoH recommended all PLHIV receive TPT regardless of ART status. Likewise, lines 143-144 states ‘the same national eligibility criteria for TPT initiation before the campaign, were maintained.’ But based on what’s written in the discussion, eligibility was expanded from only those newly enrolled in care to all ART clients, correct? Would suggest reviewing these statements for consistency.

- Lines 177-179 state that PLHIV initiated on TPT were provided with monthly refills and follow-up until they had completed 6 months of treatment. However, lines 127-129 state that TPT refill visits are aligned with ART refill visits and are given for either 1 month, 3 months, or 6 months. Would suggest reviewing these statements for consistency.

Reviewer #4: (No Response)

Reviewer #5: General comments: I have some questions and suggestions for this manuscript that will hopefully improve the communication of how analyses were implemented and improve the reader's understanding.

As a non-statistical comment, it would be really interesting to see one more year of data to see if these gains were sustained.

Specific comments:

1. (lines 223-242) The experimental unit of these analyses is unclear to me. Did you analyze data at the individual level or aggregated numbers from facilities? Said differently, is your outcome in these models a binary yes/no of TPT uptake or TPT completion or is the number of TPT uptakes or TPT completions at the facility (hopefully with an offset included in the regression model)? Based on lines 235-237, I am pretty sure this is an individual-level model, but it would be good to make this explicit.

2. (Ignore if individual-level model) If you have a count-based model, did you check for overdispersion? Overdispersion can be a big problem in Poisson regression and I strongly urge the authors to move a negative binomial model or demonstrate that there is no overdispersion.

3. I am curious about the choice of Poisson over log binomial for binary outcomes. It is possible and, if done correctly (which I think you have), acceptable, e.g., https://doi.org/10.1093/aje/kwh090, but I am curious about the choice. I think my confusion in the previous couple comments centers around that choice.

4. (lines 227-230) Please provide a methodological citation for GEE.

5. (lines 230-231) This statement is incorrect. The odds ratio does not overstate the association. If it did, there was no way it would be well accepted. What you mean to say is that the odds ratio overestimates the *risk* in outcomes that are prevalent, which is a known fact. That is due to flaws in how people interpret the results, not the method itself.

6. (lines 232-233) This seems redundant with the previous sentence. Maybe I am missing something?

7. (lines 243-245) Please provide methodological citations for all methods used here, e.g., VIFs, MICE, etc.

8. (Figures 2-5) Please describe in the methods section how the predicted probabilities are computed.

9. (Figures 2-5) It would also be good to shift the points slightly in order to better see the confidence interval overlap. For instance, in Figure 4a, it's impossible to tell which CIs are overlapping. Unfortunately, I don't know how to do this in Stata. In R's ggplot package, this can be done with the position_dodge function.

7. PLOS authors have the option to publish the peer review history of their article (what does this mean?). If published, this will include your full peer review and any attached files.

Reviewer #1: **Yes: **Peter MacPherson

Reviewer #2: No

Reviewer #3: No

Reviewer #4: No

Reviewer #5: No

---

## [Author Response · Author response to Decision Letter 2]

19 Jan 2023

The Editor, 

PLoS One 

Date: 19th January 2023

Dear Sir, 

RE: RESPONSE TO REVIEWERS’ COMMENTS MANUSCRIPT ID: PONE-D-22-13680R2 “Increased uptake of tuberculosis preventive therapy (TPT) among people living with HIV following the 100-days accelerated campaign: A retrospective review of routinely collected data at six urban public health facilities in Uganda”. 

The authors of the reviewed manuscript thank the academic editor and reviewers for their consideration and careful review of the manuscript. Below, we provide detailed point-by-point responses to the reviewers’ comments in maroon font and italics. We have also tracked the changes in the revised manuscript.

 In this document, we have also provided page and line numbers as referenced from the tracked manuscript to enable reviewers easily track the changes made as per their comments. 

Additional Editor Comments:

Many thanks for the first revisions. As seen below, Reviewer 3 has still some critical concerns around the analysis.

6. Review Comments to the Author

Reviewer #1: Thanks to the authors for a comprehensive response, which has strengthened the manuscript. I have no other questions.

Response: Thank you.

Reviewer #2: The authors have responded to all of the queries. I have two final thoughts:

- Line 158: The authors can remove "arbitrarily," which has a negative connotation. There is rationale for the starting point of the evaluation.

Response: Thank you for the comment.

The word “arbitrarily” has been removed. See page 9, line 168.

- Analysis/Discussion: The TPT completion rates are extremely high. Though promising, I worry this could be partially due to bias from only using the EMR records (which comprised only approximately 30% of the TPT initiation from 2019). The authors address this is the discussion, but conclude that there was unlikely to be bias. I am less certain, and think the results warrant a bit more caution in their interpretation.

Response: Thank you for the comment.

The statement referred to in the discussion under limitations has been removed. See page 28, line 557-559. 

Reviewer #3: Thank you for the opportunity to review this revised manuscript. The authors present findings from an evaluation of a TPT uptake campaign in Uganda. Further revisions are necessary before I would recommend this manuscript for publication.

Major comments:

- Given that the short duration of the campaign and the focus on completion for patients initiated in Q4 2018 and Q1 2019, I would suggest that the primary analysis be conducted quarterly rather than annually (as opposed to the quarterly data being presented only in the appendix). Or is there a statistical reason to focus on annual data for the primary analysis?

Response: Thank you for the comment.

We agree with your recommendation.

The analysis has been repeated and presented quarterly data as the primary analysis.

- On lines 392-394 of the discussion, the authors indicate that national guidance before 2018 was to enroll only the newly enrolled PLHIV in care and TPT eligibility was expanded in 2019. In figure 1, it’s indicated that 38,704 patients were eligible for TPT in 2016, for example. Is this correct? Are these only newly enrolled? If this is all PLHIV, I would suggest that the primary analysis be focused only on newly enrolled PLHIV since they were the only patients eligible until the last year covered by this analysis. Data on established ART clients could be presented in a supplementary appendix.

- Figure 1 and table 1 have slightly different numbers – as per the footnote, table 1 excludes individuals who were in paper based registers but not the EMR. Can figure 1 be revised so that the data across figure 1 and table 1 are consistent?

Response: Thank you for the comment.

According to the national TPT guidelines, all PLHIV in care were eligible for TPT (See ref#14). However, in instances when there was TPT shortages at the health facilities, the national guidance to the health facilities was to prioritize TPT initiation to the PLHIV not yet on ART due to their vulnerability in terms of acquiring active TB compared the ART established clients. Nevertheless, all PLHIV could be initiated on TPT if sufficient stock was available. Between 2018 and 2019, there was an increased TPT supplies at the national level, and such shortages were minimized. This provides the reason for higher TPT uptake among those “not on ART” compared to the “ART established clients” before 2018, which was leveled after the 100-day accelerated TPT campaign. We have edited the statement in the discussion to read better and have consistency. See page#26, lines 505-510.

 Figure 1 has been revised so that the data are consistent with that in table 1. On fig 1, we have inserted another layer (i.e. Eligible & analyzed for TPT uptake) to show the numbers patients actually analyzed for TPT uptake per year after excluding those who appeared in TPT paper-based register but had no information the EMR. See pages #14-15.

Minor comments:

- On lines 89-91 (and again on lines 136-137), the authors describe the 100 day campaign and indicate that completion was focused on patients who initiated in the quarters October-December 2018 and January-February 2019. Should the second quarter be January-March 2019 (to reflect a 3 month period?)?

Response: Thank you for the comment.

We agree, the second quarter targeted for 100% TPT completion in the 100-day should be January-March 2019 to reflect the 3 months period. This was however stated as January-February 2019 in the referenced document – 100-Day Accelerated Isoniazid Preventive Therapy Scale Up Plan (ref#18).

To make the quarter complete, we have edited the statements to indicate it as January-March 2019. See page #6, line# 101, page#8, line# 147, and page# 25, lines 485 – 489.

- On line 110, the comma after ‘inpatient health’ should come after the word ‘services’

Response: Thank you for the comment.

The comma has been moved. See page# 7, line 120.

- Line 256 states that 95% of patients were eligible for TPT during each year. However, in 2018, it was 91% (40,390/44,439) and in 2019, it was 89% (40,867/45,868). Line 256 should be revised to reflect the data presented in figure 1.

Response: Thank you for the comment.

The statement has been revised with correct value to match data presented in fig1. See page #14, lines 283.

- On lines 258-259, the authors state that ‘data were missing on 10% and 4% of patients for TPT uptake and completion analyses, respectively.’ However, 26,478 people initiated TPT in 2019 and only 7,713 were evaluated for completion. So isn’t completion data missing for 71% of individuals who initiated TPT in that year?

Response: Thank you.

A statement has been added to show the 70% data missingness on TPT outcomes for individuals excluded from TPT completion analysis due to the unmatched IDs from the electronic HIV care database and paper-based TPT registers. See page #14, lines #284 – 289.

- In the response to reviewer #2 (point 4), the authors indicated that the decline observed in 2018 was due to stockouts at the national medical stores but I didn’t see this reported in the revised manuscript. This could be added to the discussion to provide more context for the results.

Response: Thank you for the comment.

A statement on the cause of the TPT uptake decline in 2018 has been added in the discussion section. See page #25, lines #489 – 493.

- While the discussion states that national guidance was to provide TPT to only those newly enrolled in care through 2018, the introduction (lines 77-79) and the methods (lines 120-122) state that the MoH recommended all PLHIV receive TPT regardless of ART status. Likewise, lines 143-144 states ‘the same national eligibility criteria for TPT initiation before the campaign, were maintained.’ But based on what’s written in the discussion, eligibility was expanded from only those newly enrolled in care to all ART clients, correct? Would suggest reviewing these statements for consistency.

Response: Thank you for the comment.

The statement in discussion has been revised to have consistency. 

In the program setting, the guidance to health facilities providing TPT was to prioritize TPT for newly enrolled in care “not on ART” in situations of drug stock-outs. However, all PLHIV were eligible and could be initiated on TPT when drugs were available. See page#26, lines 506-513.

- Lines 177-179 state that PLHIV initiated on TPT were provided with monthly refills and follow-up until they had completed 6 months of treatment. However, lines 127-129 state that TPT refill visits are aligned with ART refill visits and are given for either 1 month, 3 months, or 6 months. Would suggest reviewing these statements for consistency.

Response: Thank you for the comment.

The monthly TPT refills and follow-up until completion of a 6-months treatment course was the guideline, however, in a programmatic setting, the practice is to provide TPT refills on the same day of ART refill visit. This helps to achieve TPT adherence and also reduces the number of times an individual has to visit the clinic which can be costly and time consuming.

A statement with additional explanation has been added in the treatment procedure section. See page #9, lines #187 – 188, and page #10, lines #189 – 192.

Reviewer #4: (No Response)

Response: Thank you.

Reviewer #5: General comments: I have some questions and suggestions for this manuscript that will hopefully improve the communication of how analyses were implemented and improve the reader's understanding.

As a non-statistical comment, it would be really interesting to see one more year of data to see if these gains were sustained.

Response: Thank you for the comment.

We agree, it would be interesting if more years of data are added to examine if these gains were sustained. However, the objective of this manuscript was to evaluate whether the 100-days of accelerated campaign achieved its objectives: (1) increasing TPT uptake from about 30% to about 50% during the 100-days of accelerated TPT initiation i.e. from July 2019 to October 2019. (2) achieve 100% completion until September 2019 among PLHIV initiated on TPT during the quarters of October - December 2018 and January – March 2019. 

We have added a statement recommending future evaluation to assess sustainability of these gains with more years of data post the 100-days accelerated TPT intervention. See page #29, lines #576 – 578.

Specific comments:

1. (lines 223-242) The experimental unit of these analyses is unclear to me. Did you analyze data at the individual level or aggregated numbers from facilities? Said differently, is your outcome in these models a binary yes/no of TPT uptake or TPT completion or is the number of TPT uptakes or TPT completions at the facility (hopefully with an offset included in the regression model)? Based on lines 235-237, I am pretty sure this is an individual-level model, but it would be good to make this explicit.

Response: Thank you for the comment.

Data was analyzed on individual level not aggregated numbers. TPT uptake or TPT completion were entered as binary (yes/no) outcomes when fitting the Poisson models with robust error variance. 

A statement has been added in statistical methods section to make it clear the unit of analysis. See page #12, lines #241 – 242.

2. (Ignore if individual-level model) If you have a count-based model, did you check for overdispersion? Overdispersion can be a big problem in Poisson regression and I strongly urge the authors to move a negative binomial model or demonstrate that there is no overdispersion.

Response: Thank you for the comment.

Data was analyzed on individual level not aggregated as indicated in response to the previous comment.

3. I am curious about the choice of Poisson over log binomial for binary outcomes. It is possible and, if done correctly (which I think you have), acceptable, e.g., https://doi.org/10.1093/aje/kwh090, but I am curious about the choice. I think my confusion in the previous couple comments centers around that choice.

Response: Thank you for the comment.

The choice of Poisson regression with robust variance estimates, over log-binomial is because the former directly and appropriately estimates relative risk whereas the latter often face a convergence problem which requires proper choice of starting values.

A statement on justification for the choice of the model used has been added. See page #12, lines #257 – 259.

4. (lines 227-230) Please provide a methodological citation for GEE.

Response: Thank you for the comment.

A citation for GEE has been provided (reference#26). See page #12, lines #248 – 251.

5. (lines 230-231) This statement is incorrect. The odds ratio does not overstate the association. If it did, there was no way it would be well accepted. What you mean to say is that the odds ratio overestimates the *risk* in outcomes that are prevalent, which is a known fact. That is due to flaws in how people interpret the results, not the method itself.

Response: Thank you for the comment.

The statement has been edited. See page #12, lines #252 – 253.

6. (lines 232-233) This seems redundant with the previous sentence. Maybe I am missing something?

Response: Thank you for the comment.

The statement referred to in this comment explains the model used for examining factors associated with TPT completion, while the previous statement in the same section explains the model for examining factors associated with TPT uptake. See page #12, lines #248 – 257.

7. (lines 243-245) Please provide methodological citations for all methods used here, e.g., VIFs, MICE, etc.

Response: Thank you for the comment.

Citations have been provided. VIFs (reference#29), MICE (reference#30), and best-worst case scenario (reference #31). See page #13, lines #270 – 272.

8. (Figures 2-5) Please describe in the methods section how the predicted probabilities are computed.

Response: Thank you for the comment.

A statement has been added on how the predicted probabilities were computed. See page #13, lines #267 – 269.

9. (Figures 2-5) It would also be good to shift the points slightly in order to better see the confidence interval overlap. For instance, in Figure 4a, it's impossible to tell which CIs are overlapping. Unfortunately, I don't know how to do this in Stata. In R's ggplot package, this can be done with the position_dodge function.

Response: Thank you for the comment.

Plots have been redone in “R” using position_dodge function to shift points a bit and be able to see confidence interval overlap. See pages # 31-38.

---

## [Decision Letter · Decision Letter 3]

31 Jan 2023

PONE-D-22-13680R3Increased uptake of tuberculosis preventive therapy (TPT) among people living with HIV following the 100-days accelerated campaign: A retrospective review of routinely collected data at six urban public health facilities in UgandaPLOS ONE

Dear Dr. Musaazi,

Thank you for submitting your manuscript to PLOS ONE. After careful consideration, we feel that it has merit but does not fully meet PLOS ONE’s publication criteria as it currently stands. Therefore, we invite you to submit a revised version of the manuscript that addresses the points raised during the review process.

 There is one small comment from Reviewer 3 left (description of the 100 day campaign). Please address as soon as possible. Once addressed, an editorial decision will be taken (the manuscript will not be re-reviewed).

We look forward to receiving your revised manuscript.

Kind regards,

Lukas Fenner, MD, MSc

Academic Editor

PLOS ONE

Journal Requirements:

Reviewers' comments:

Reviewer's Responses to Questions

**Comments to the Author**

1. If the authors have adequately addressed your comments raised in a previous round of review and you feel that this manuscript is now acceptable for publication, you may indicate that here to bypass the “Comments to the Author” section, enter your conflict of interest statement in the “Confidential to Editor” section, and submit your "Accept" recommendation.

Reviewer #3: All comments have been addressed

Reviewer #5: All comments have been addressed

2. Is the manuscript technically sound, and do the data support the conclusions?

Reviewer #3: Yes

Reviewer #5: (No Response)

3. Has the statistical analysis been performed appropriately and rigorously? 

Reviewer #3: I Don't Know

Reviewer #5: (No Response)

4. Have the authors made all data underlying the findings in their manuscript fully available?

Reviewer #3: Yes

Reviewer #5: (No Response)

5. Is the manuscript presented in an intelligible fashion and written in standard English?

Reviewer #3: Yes

Reviewer #5: (No Response)

6. Review Comments to the Author

Reviewer #3: Thanks for the revisions to your paper. I have one last comment (please note, it is not necessary for me to re-review this paper in light of this comment):

In my previous review, I had noted that the 100 day campaign was described as focusing on patients who initiated in quarters Oct-Dec 2018 and Jan-Feb 2019 and asked whether the second quarter should reflect Jan-Mar 2019. In the authors reply, they indicated that they had changed the text to reflect a full quarter but that the 100 day campaign reference document was focused on Jan-Feb 2019. I would recommend describing the campaign as it was implemented. If the focus was truly on Jan-Feb 2019 patients, I would state that so that your manuscript is accurately reflecting the Ministry of Health’s program.

Reviewer #5: (No Response)

7. PLOS authors have the option to publish the peer review history of their article (what does this mean?). If published, this will include your full peer review and any attached files.

Reviewer #3: No

Reviewer #5: No

---

## [Author Response · Author response to Decision Letter 3]

1 Feb 2023

The Editor, 

PLoS One 

Date: 01st February 2023

Dear Sir, 

RE: RESPONSE TO REVIEWERS’ COMMENTS MANUSCRIPT ID: PONE-D-22-13680R3 “Increased uptake of tuberculosis preventive therapy (TPT) among people living with HIV following the 100-days accelerated campaign: A retrospective review of routinely collected data at six urban public health facilities in Uganda”. 

The authors of the reviewed manuscript thank the academic editor and reviewers for their consideration and careful review of the manuscript. Below, we provide detailed point-by-point responses to the reviewers’ comments in maroon font and italics. We have also tracked the changes in the revised manuscript.

 In this document, we have also provided page and line numbers as referenced from the tracked manuscript to enable reviewers easily track the changes made as per their comments. 

Additional Editor Comments:

Thank you for submitting your manuscript to PLOS ONE. After careful consideration, we feel that it has merit but does not fully meet PLOS ONE’s publication criteria as it currently stands. Therefore, we invite you to submit a revised version of the manuscript that addresses the points raised during the review process.

There is one small comment from Reviewer 3 left (description of the 100 day campaign). Please address as soon as possible. Once addressed, an editorial decision will be taken (the manuscript will not be re-reviewed).

6. Review Comments to the Author

Reviewer #3: Thanks for the revisions to your paper. I have one last comment (please note, it is not necessary for me to re-review this paper in light of this comment):

In my previous review, I had noted that the 100 day campaign was described as focusing on patients who initiated in quarters Oct-Dec 2018 and Jan-Feb 2019 and asked whether the second quarter should reflect Jan-Mar 2019. In the authors reply, they indicated that they had changed the text to reflect a full quarter but that the 100 day campaign reference document was focused on Jan-Feb 2019. I would recommend describing the campaign as it was implemented. If the focus was truly on Jan-Feb 2019 patients, I would state that so that your manuscript is accurately reflecting the Ministry of Health’s program.

Response: Thank you for the comment.

We have made edits and indicated the period the 100-day campaign targeted for TPT completion (i.e. initiated in quarters Oct-Dec 2018 and Jan-Feb 2019) as stated in the 100-day campaign reference document.

See page #6 line 95, page #8 line 141, and page #24 line 465.

---

## [Editor Report · Decision Letter 4]

7 Feb 2023

Increased uptake of tuberculosis preventive therapy (TPT) among people living with HIV following the 100-days accelerated campaign: A retrospective review of routinely collected data at six urban public health facilities in Uganda

PONE-D-22-13680R4

Dear Dr. Musaazi,

We’re pleased to inform you that your manuscript has been judged scientifically suitable for publication and will be formally accepted for publication once it meets all outstanding technical requirements.

Kind regards,

Lukas Fenner, MD, MSc

Academic Editor

PLOS ONE
---

## [Editor Report · Acceptance letter]

10 Feb 2023

PONE-D-22-13680R4 

Increased uptake of tuberculosis preventive therapy (TPT) among people living with HIV following the 100-days accelerated campaign: A retrospective review of routinely collected data at six urban public health facilities in Uganda 

Dear Dr. Musaazi:

I'm pleased to inform you that your manuscript has been deemed suitable for publication in PLOS ONE. Congratulations! Your manuscript is now with our production department. 

Kind regards, 

on behalf of

Prof. Lukas Fenner 

Academic Editor

PLOS ONE